

# Coupled dynamics and evolution of primordial and recycled heterogeneity in Earth's lower mantle

Anna Johanna Pia Gülcher[1], Maxim Dyonis Ballmer[2,1], and Paul James Tackley[1]

[1]Institute of Geophysics, Department of Earth Sciences, ETH Zürich, Zürich, Switzerland
[2]Department of Earth Sciences, University College London, London, UK

**Correspondence:** Anna J. P. Gülcher (anna.guelcher@erdw.ethz.ch)

**Abstract.** The nature of compositional heterogeneity in Earth's lower mantle remains a long-standing puzzle that can inform about the long-term thermochemical evolution and dynamics of our planet. Here, we use global-scale 2D models of thermo-chemical mantle convection to investigate the coupled evolution and mixing of (intrinsically-dense) recycled and (intrinsically-strong) primordial heterogeneity in the mantle. We explore the effects of ancient compositional layering of the mantle, as

motivated by magma-ocean solidification studies, and of the physical parameters of primordial material. Depending on these physical parameters, our models predict various regimes of mantle evolution and heterogeneity preservation over 4.5 Gyrs. Over a wide parameter range, primordial and recycled heterogeneity are predicted to co-exist with each other in the lower mantle of Earth-like planets. Primordial material usually survives as mid-to-large scale blobs (or streaks) in the mid-mantle, around 1000-2000 km depth. This preservation is largely independent on the initial primordial-material volume. In turn, recy-

cled oceanic crust (ROC) persists as large piles at the base of the mantle and as small streaks everywhere else. In models with a dense FeO-rich layer that is initially present at the base of the mantle, the FeO-rich material partially survives at the top of ROC piles, causing the piles to be compositionally stratified. Moreover, the addition of an ancient FeO-rich basal layer in the lowermost mantle significantly aids the preservation of the viscous domains in the mid-mantle. Primordial blobs are commonly (but not always) directly underlain by thick ROC piles, and aid their longevity and stability. The preservation of primordial

domains along with recycled piles is relevant for Earth as it may reconcile geophysical and geochemical constraints on lower mantle heterogeneity.


## 1 Introduction

The lower mantle is the largest geochemical reservoir in the Earth's interior, and controls the style of mantle convection and
planetary evolution. Despite efficient stirring by vigorous convection over billions of years, the Earth's lower mantle appears
to be chemically heterogeneous on various length scales (e.g., Allegre and Turcotte, 1986; van Keken and Ballentine, 1998;
Waszek et al., 2018). Constraining the distribution of this heterogeneity is key for assessing Earth's bulk composition and
thermochemical evolution, but remains a scientific challenge that requires cross-disciplinary efforts.

On relatively small scales (mm to km), the concept of a "marble cake" mantle has gained wide acceptance, emphasising
that much of the mantle is made out of recycled oceanic lithosphere, deformed into narrow streaks of depleted and enriched
compositions (Fig. 1a). Partial melting in the upper mantle creates heterogeneity between basaltic (magma) and harzburgitic
(residue) end-members, forming a physically and chemically layered oceanic lithosphere. Subsequent injection into the man-
tle during subduction causes a non-equilibrated mantle that is a mechanical mixture of basalt and harzburgite (Allegre and
Turcotte, 1986; Christensen and Hofmann, 1994; Morgan and Morgan, 1999; Xu et al., 2008; Ritsema et al., 2011). In addi-
tion, small-scale heterogeneity is introduced into the mantle by recycling of continental material through subduction and/or
delamination. Such introduction of felsic material in the deeper Earth has been used to explain the heterogeneous isotopic
compositions of mantle-derived rocks, in particular in terms of trace elements (Hofmann, 1997; Kawai et al., 2009; Stracke,
2012).

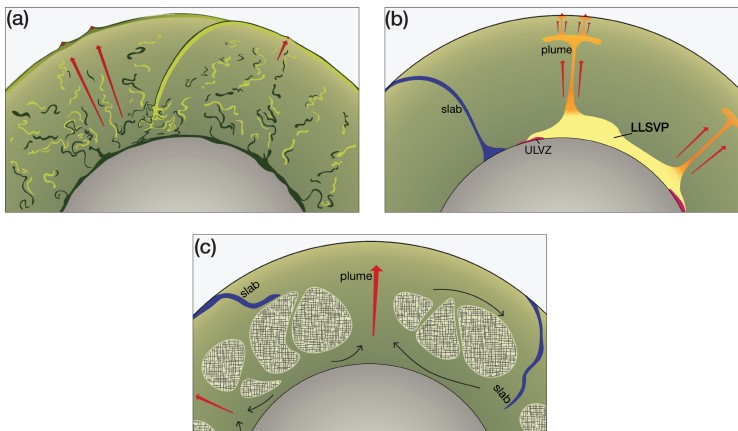

**Figure 1.** Several conceptual models of mantle compositional structure with heterogeneity on various scales. For explanations regarding
these heterogeneity styles, the reader is referred to the text. **a) "Marble cake" mantle**, modified after Xu et al. (2008); Woodhead (2015);
**b) Thermochemical piles**, modified after Deschamps et al. (2015); **c) Viscous "blobs"**, modified after Becker et al. (1999); Ballmer et al.
(2017a).



On larger scales (10s-100s of km), compositional heterogeneity may be preserved by delayed mixing of this marble cake
with to either intrinsically-dense or intrinsically-strong materials. Intrinsically dense materials may accumulate as piles atop
the core-mantle boundary (Fig. 1b). In particular, the two large low-shear velocity provinces (LLSVPs) in the deep Earth have
likely resisted mantle mixing due to their thermochemical origin, with an excess density of a few percent compared to pyrolitic
material (Hernlund and Houser, 2008). Hypotheses regarding the composition of these piles involve recycled oceanic crust
(ROC) (e.g., Christensen and Hofmann, 1994; Hirose et al., 2005; Li et al., 2014a), ancient dense material (e.g., Labrosse et al.,
2007; Tackley, 2012; Bower et al., 2013; Li et al., 2014b) or a combination of the two (e.g., Tackley, 2012; Ballmer et al.,
2016).

On the other hand, intrinsically viscous domains may survive mantle convection and be preserved as "blobs" in the mid-
mantle for large timescales (Manga, 1996; Becker et al., 1999; Ballmer et al., 2017a; Gülcher et al., 2020b), such as plums
in the mantle "plum pudding" (Fig. 1c). The physical properties (e.g., high viscosity) required for long-term preservation of
these blobs are thought to be caused by an enrichment in the strong lower-mantle mineral bridgmanite $(Mg,Fe)SiO_3$ (i.e.,
stabilised by an enrichment in silica). During crystallization of a deep magma ocean at high pressures, $MgSiO_3$ bridgmanite is
the relevant liquidus phase over a wide range of conditions and may hence be fractionated, potentially giving rise to the such
ancient heterogeneity in the mantle (Elkins-Tanton, 2008; Boukaré et al., 2015; Xie et al., 2020). These viscous domains in
the mid-mantle may potentially reconcile recent seismic observations of mid-mantle heterogeneity (e.g., Fukao and Obayashi,
2013; Jenkins et al., 2017; Waszek et al., 2018) as well as cosmochemical and geochemical constraints that indicate that the
lower mantle hosts an ancient primordial reservoir, potentially enriched in $SiO_2$, with respect to the upper mantle (e.g., Jackson
et al., 2010; Mukhopadhyay, 2012; Touboul et al., 2012). This regime has been successfully reproduced in 2D spherical annulus
convection models with composition-dependent rheology (Gülcher et al., 2020b). Various styles of primordial heterogeneity
preservation were found as a function of its physical parameters, and the survival of sharp-to-diffuse primordial domains in
the mid-mantle was proposed for planet Earth (Gülcher et al., 2020b). Yet, the employed rheology was simplified and the
models did not produce Earth-like tectonic behaviour. Moreover, none of the models explicitly predicted the formation of
thermochemical piles in the lowermost mantle, which are perhaps the most evident heterogeneities in Earth's lower mantle
(e.g., Solomatov and Stevenson, 1993; Christensen and Hofmann, 1994; Li and Romanowicz, 1996).

Many studies have explored the formation and preservation of either ROC (intrinsically dense) or primordial (intrinsically
dense and/or viscous) heterogeneity, but only few if any have quantified mantle dynamics in the presence of different types
of heterogeneity with distinct physical properties (e.g., Nakagawa and Tackley, 2014; Li et al., 2014b). Understanding the
interplay between recycled and primordial heterogeneity is critical to validate, falsify, and/or integrate the various views on
chemical heterogeneity in Earth's mantle (Fig. 1). The goal of the present contribution is to investigate mantle dynamics in
the presence of different types of heterogeneity with distinct physical properties (i.e. intrinsically viscous and dense), using
Earth-like numerical models of mantle convection. We establish multiple regimes of chemical heterogeneity, dependent on
the rheological properties of primordial material. The coexistence of viscous, primordial blobs in the mid-mantle with dense
recycled (and possible ancient) piles in the lowermost mantle is robustly predicted in many experiments. This coexistence is
largely independent of the initial compositional layering set-up. We propose a new style of chemical heterogeneity in Earth's



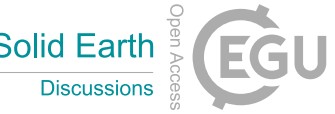

mantle, that is in a hybrid state between "marble cake" and "plum pudding" styles. Finally, the results are put into context with
recent discoveries from geochemistry and seismology, and the applicability to the evolution of Earth.

## 2    Methods

### 2.1    Numerical technique and initial set-up

In this study, we use the finite-volume code StagYY (Tackley, 2008) to model mantle convection in two-dimensional spherical
annulus geometry (Hernlund and Tackley, 2008). The conservation equations for mass, momentum, energy and composition
are solved on a staggered grid for a compressible fluid with an infinite Prandtl number. The modelled mantle domain is dis-
cretised by 512·96 cells. Due to the spherical geometry, as well as vertical grid refinement near the boundary layers and near
660 km depth, the size of grid cells varies between 10 and 35 km in the vertical, and 40 and 80 km in the horizontal direction,
respectively. Around 1.2 million tracers (25 tracers per cell) are used to handle non-diffusive advection of composition and
temperature. Boundary conditions are free-slip and isothermal at the top and bottom boundaries, achieved by imposing temper-
atures of 300 K and 4000 K, respectively. The numerical experiments are purely bottom-heated (no internal heating). The initial
temperature profile involves an adiabat with a potential temperature of 1900 K, mimicking increased mantle temperatures in
the early Earth (e.g., Herzberg et al., 2010; Herzberg and Rudnick, 2012).

The initial condition of composition in our models are simplified two-layered (and for selected cases, three-layered) profiles
motivated by a fractional-crystallization sequence of the magma ocean (Elkins-Tanton, 2008; Boukaré et al., 2015; Xie et al.,
2020). In the lower mantle, bridgmanite is the liquidus phase for a wide range of compositions, while the rest of the mantle
evolves towards pyrolitic compositions as bridgmanite is fractionated (Ito and Takahashi, 1989). Similar to previous work
(Gülcher et al., 2020b), we impose a bridgmanitic "primordial" material layer in the lower mantle with layer thickness $D_{\mathrm{prim}}$
(2230 km for the reference case). This primordial layer includes 5% pyrolitic "noise", distributed randomly throughout the
layer, resulting in an initial primordial layer that is not a pristine fractional-crystallization end-member cumulate. The upper
mantle is initially a mechanical mixture of 15% basalt and 85% harzburgite, i.e. close to pyrolitic composition (see Appendix
A1).

### 2.2    Rheology

We apply a visco-plastic rheology, assuming that the material deforms plastically once a critical pressure-dependent yield stress
is reached (as in Tackley, 2000; Crameri and Tackley, 2014). Physical and rheological parameters used in this study are listed in
Table 1. Viscous deformation is governed by a simplified temperature-, pressure- and composition-dependent Arrhenius-type
viscosity law (Newtonian rheology):

$$\eta(T, P, c) = \eta_0 \cdot \lambda_{\mathrm{c}} \cdot exp\left(\frac{E_{\mathrm{a}} + P \cdot V_{\mathrm{a}}}{R \cdot T} - \frac{E_{\mathrm{a}}}{R \cdot T_0}\right) \tag{1}$$

where $\eta_0$ is the reference viscosity at zero pressure and reference temperature $T_0$ (=1600 K), $E_{\mathrm{a}}$ is the activation energy, $V_{\mathrm{a}}$
is the activation volume, $T$ the absolute temperature, $P$ the pressure and $R$ is the gas constant (8.314 J·mol$^{-1}$K$^{-1}$). As one





**Table 1.** Physical properties used in the simulations of this study. UM = upper mantle, PV = perovskite, PPV = postperovskite. Since we solve for compressible convection, the adiabatic temperature, density, thermal conductivity, thermal expansivity, and heat capacity are pressure-dependent following a third-order Birch-Murnaghan equation of state (Tackley et al., 2013).

| Property | Symbol | Value | Units |
|---|---|---|---|
| Mantle domain thickness | $D$ | 2890 | km |
| Primordial layer thickness* | $D_{prim}$ | 1700-2230 | km |
| Gravitational acceleration | $g$ | 9.81 | m/s$^3$ |
| Surface temperature | $T_s$ | 300 | K |
| CMB temperature | $T_{CMB}$ | 4000 | K |
| Reference viscosity | $\eta_0$ | $5\cdot10^{20}$ | Pa·s |
| Reference lower-mantle viscosity contrast* | $\lambda_{LM}$ | 1 | |
| Primordial lower-mantle viscosity contrast* | $\lambda_{prim}$ | 1-500 | |
| PPV viscosity contrast | $\lambda_{ppv}$ | $10^{-3}$ | |
| Primordial buoyancy ratio* | $B$ | 0.07-0.78 | |
| Reference temperature | $T_0$ | 1600 | K |
| Initial reference temperature | $T_{0,ini}$ | 1900 | K |
| Activation energy - UM & PV | $E_{a,UM}$ | 140 | kJ/mol |
| Activation energy - PPV | $E_{a,PPV}$ | 100 | kJ/mol |
| Activation volume - UM & PV | $V_{a,UM}$ | $1.8\cdot10^{-6}$ | cm$^3$/mol |
| Activation volume - PPV | $V_{a,PPV}$ | $1.4\cdot10^{-6}$ | cm$^3$/mol |
| Yield stress | $\tau_{yield}$ | 30 | Mpa |
| Yield stress depth derivative | $\tau'_{yield}$ | 0.01 | MPa/MPa |
| Surface specific heat capacity | $C_{P,0}$ | 1200 | J/(kg·K) |
| Surface thermal conductivity | $k_0$ | 3 | W/(m·K) |
| Surface thermal expansivity | $\alpha_0$ | $3\cdot10^{-5}$ | K$^1$ |

of the main model ingredients, we consider the composition-dependency of viscosity through prefactor $\lambda_c$ that can be layer dependent (see Section 2.4). Our relatively low effective activation energy may represent the thermodynamic properties of lower-mantle materials, or mimic the effects of a complex rheology dependent on grain size or stress (e.g., Yang and Gurnis, 2016; Glišovic et al., 2015).

We determine models with plate-like behaviour using the same diagnostics as in Tackley 2000, measuring plateness $P$ (the degree to which surface deformation is localized) and mobility $M$ (the extent to which the lithosphere is able to move). For details regarding these diagnostics, the reader is referred to Appendix B. Plate-like behaviour occurs for $P$ close to 1 and $M$ close to or larger than 1 Tackley (2000).





**Table 2.** Phase change parameters used in this study for the olivine, pyroxene-garnet, and primordial system (the latter is parametrised to fit the density profile of a mixture of 40% basalt and 60% harzburgite from Xu et al. 2008). The table shows the depth and temperature at which a phase transition occurs; $\triangle\rho_{\mathrm{pc}}$ and $\gamma$ denote the density jump across the phase transition and the Clapeyron slope, respectively. Phase change parameters used for the olivine and pyroxene-garnet systems are similar to previous studies (e.g., Tackley et al., 2013). Finally, $K_0$ refers to the reference bulk modulus for the system for each individual layer (marked by the depth range), and is increased to 230 GPa for primordial material in the lower mantle (Wolf et al., 2015).

| Depth [km] | Temperature [K] | $\triangle\rho$ [kg/m$^3$] | Phase change width [km] | $\gamma$ [MPa/K] | $K_0$ [GPa]; depth range [km] |
|---|---|---|---|---|---|
| *Olivine ($\rho_{\mathrm{surf}}$=3240 kg/m$^3$)* | | | | | 163; 0-410 |
| 410 | 1600 | 180 | 25 | +2.5 | 85; 410-660 |
| 660 | 1900 | 435 | 25 | -2.5 | 210; 660-2740 |
| 2740 | 2300 | 61.6 | 25 | +10 | 210; 2740-2890 |
| *Pyroxene-garnet ($\rho_{\mathrm{surf}}$=3080 kg/m$^3$)* | | | | | 163; 0-40 |
| 40 | 1000 | 350 | 25 | 0 | 130; 40-300 |
| 300 | 1600 | 100 | 75 | +1.0 | 85; 300-720 |
| 720 | 1900 | 350 | 75 | +1.0 | 210; 720-2740 |
| 2740 | 2300 | 61.6 | 25 | +10 | 210; 2740-2890 |
| *Primordial ($\rho_{\mathrm{surf}}$=3075 kg/m$^3$)* | | | | | 163; 0-40 |
| 40 | 1000 | 260 | 25 | 0 | 145; 40-380 |
| 380 | 1600 | 130 | 75 | 1.675 | 85; 380-660 |
| 660 | 1900 | 450 | 50 | -0.575 | 230; 660-2740 |
| 2740 | 2300 | 61.6 | 25 | 10 | 163; 0-410 |

### 2.3 Mantle composition, phase changes and melting

We consider a simplified mantle composition with three lithological components: harzburgite, basalt and primordial material.
Accordingly, each tracer carries either a primordial material composition or a mechanical mixture of harzburgite and basalt (hz and bs). The initial composition in the upper mantle is a mechanical mixture of 85% hz and 15% bs, i.e. slightly depleted relative to present-day pyrolitic material (80% hz and 20% bs). Harzburgitic and basaltic materials are treated as a mixture of olivine and pyroxene-garnet systems that undergo different solid-solid phase transitions (for details, see e.g. Nakagawa et al., 2010). Primordial material is not defined in terms of a specific mineral composition, but solely through its material properties.
Physical properties and parameters for the phase transitions for each mineral system, and for primordial material, are given in Table 2.

The density profiles of the relevant mantle materials are plotted in Fig. A2. The density profile of primordial material is consistent with that of a bridgmanite-enriched material with a (Mg+Fe)/Si ratio of $\approx$1.0 (see Appendix A2). We further impose a relatively higher bulk modulus for primordial material than that of the pyrolitic mantle (230 Gpa opposed to 210
GPa), consistent with high-pressure experimental studies of bridgmanite (Wolf et al., 2015). In a recent study, such higher





bulk modulus was found to aid primordial heterogeneity preservation in the mid-mantle (Gülcher et al., 2020b). Our reference primordial material roughly corresponds to $Mg_{0.88}Fe_{0.12}SiO_2$ (Tange et al., 2012; Wolf et al., 2015), or any other material with the same density profile (see Appendix A2). The initial chemical density difference between the primordial and pyrolitic layer stabilizes chemical stratification, and its competition with the destabilizing thermal density difference is expressed as the

non-dimensional buoyancy ratio $B$ (Hansen and Yuen, 1988; Davaille, 1999):

$$B = \frac{\triangle \rho_C}{\triangle \rho_T} = \frac{\triangle \rho_C}{\rho \alpha \triangle T} \tag{2}$$

where $\triangle \rho_C$ and $\triangle \rho_T$ are the relevant compositional and thermal density contrasts; $\rho$ is the density of the lower layer; $\alpha$ is the thermal expansivity, and $\triangle T$ the super-adiabatic temperature contrast between surface and CMB. We calculate $B$ for relevant lower-mantle depths, thus taking depth-dependent parameters $\triangle \rho_C$, $\rho$ (the density of primordial material) and $\alpha$ at 1500 km

depth. Accordingly, our reference model with Mg# $=0.88$ has a buoyancy number of 0.28 (see Extended Table C1).

Compositional anomalies carried on tracers evolve from the initial state due to melt-induced differentiation. For example, tracers in the basalt-harzburgite space undergo partial melting as a function of pressure, temperature, and composition to sustain the formation of basaltic crust (for details, see Nakagawa et al. 2010). To approximate melting of primordial material, we assume that any primordial tracer is converted into a tracer with 40% basalt and 60% harzburgite once it crosses the relevant

solidus of pyroxenite melting (Pertermann and Hirschmann, 2003) (note that pyroxenes are the low-pressure polymorphs of bridgmanite). We use this 40:60 ratio as it corresponds to a (Mg+Fe)/Si ratio of $\approx$1.0 (such as in bridgmanite). This conversion flags the material as "non-primordial", since any melting and related degassing (Gonnermann and Mukhopadhyay, 2007) would likely destroy, or at least dilute, the ancient isotopic fingerprint of the previously "primordial" material. Due to this tracer conversion, the final composition of the non-primordial mantle (from here on termed "ambient mantle") $f_{bs,amb}^{final}$ evolves

over time as primordial material melts in the upper mantle:

$$f_{bs,amb}^{final} = \left(f_{bs,amb}^{ini} \cdot V_{amb}^{ini} + c_{bs,prim} \cdot V_{prim,molten}^{final}\right) / \left(V_{amb}^{ini} + V_{prim,molten}^{final}\right) \tag{3}$$

where $f_{bs,amb}^{ini}$ is the initial bulk composition of the ambient mantle with volume $V_{amb}^{ini}$, $c_{bs,prim}$ is the conversion factor of primordial material into bs-hz space (i.e. 0.4) and $V_{prim,molten}^{final}$ is the total volume of primordial material that has melted during model evolution (= $\left(1 - \chi_{prim}^{pres}\right) \cdot V_{prim}^{ini}$ with primordial preservation factor $\chi_{prim}^{pres} = \frac{V_{prim}^{final}}{V_{prim}^{ini}}$).

### 145   2.4   Parameter study

In this study we systematically run two model suites. In the first model suite, we investigate the styles of chemical heterogeneity in the mantle as a function of the physical properties of primordial material. The model parameters explored are the intrinsic density (FeO enrichment) and viscosity contrast (silica enrichment) of the primordial material relative to pyrolite (as in Ballmer et al., 2017a; Gülcher et al., 2020b). Accordingly, the density profile of primordial material is shifted throughout the mantle,

intrinsically changing the initial buoyancy ratio $B$ between the models (ranging from 0.07 to $\approx$0.78, see eq. (2) and Table C1). The approximate primordial material Mg# thereby ranges from 0.9-0.85 (see Appendix A2). Additionally, a viscosity contrast $\lambda_{prim}$ is imposed between primordial material and pyrolitic mantle material in the lower mantle, motivated by the





high viscosity of bridgmanite relative to ferropericlase (Yamazaki and Karato, 2001; Girard et al., 2016). In the second suite of models, we adjust the initial compositional layering set-up. We vary the primordial layer thickness $D_{prim}$ and for selected

cases, include a thin, dense FeO-rich layer just above the CMB (see Section 3.1.2).

## 3  Results

We have conducted 108 numerical experiments for this study. The relevant model parameters and selected output variables of each case are summarised in Extended Data Table C1. In this section, we first describe the various heterogeneity styles identified in the first model suite, after which we discuss how initial compositional layering affects mantle dynamics and

chemical heterogeneity preservation (model suite 2).

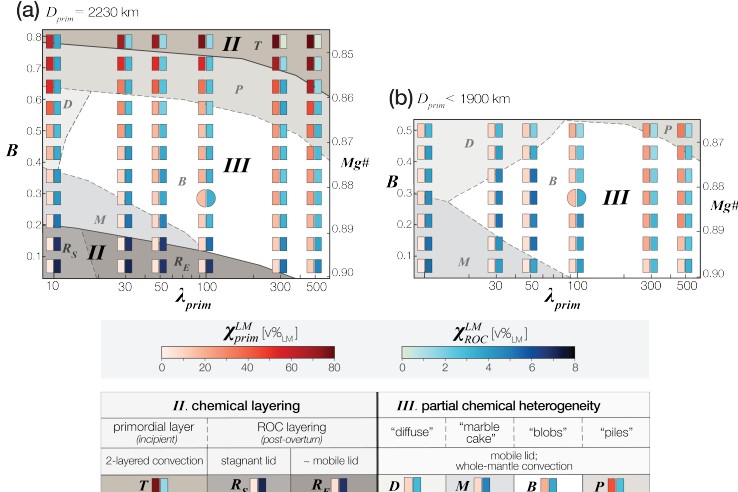

**Figure 2.** Summary of key results as a function of physical parameters of primordial material for model suites 1 **(a)** and 2 **(b)**. The vertical axis represents the initial buoyancy ratio $B$, defined at 1500 km depth (see Methods and Fig. A2), or the corresponding Mg# of primordial material. The horizontal axis gives the viscosity contrast $\lambda_{prim}$ between primordial and pyrolitic material in the lower mantle. The fraction of primordial heterogeneity ($\chi_{prim}^{LM}$, red shading) and ROC ($\chi_{ROC}^{LM}$, blue shading) in the lower mantle are shown as averages between 4.25 and 4.5 Myr model time (for definitions, see Appendix A3). The circle denotes the reference model in this study. Regime boundaries are established based on the style of mantle evolution and chemical heterogeneity (see text): (II) chemical stratification as a primordial layer with topography ("T") or post-overturn ROC layering ("R$_S$"; "R$_E$"); *(III)* partial heterogeneity preservation as marble-cake mantle ("M"), marginally stable piles ("P"), viscous blobs ("B"), or diffuse domains ("D").





## 3.1 Effect of composition-dependent rheology on mantle heterogeneity preservation

Chemical heterogeneity preservation after $\approx$4.5 Gyr model time is summarized in Fig. 2a. The results reveal two major styles of long-term convection and mixing in the mantle, which are further divided into several subregimes. We label these regimes in line with previous work on variable styles of primordial heterogeneity preservation (Gülcher et al., 2020b) as regimes *I-*

*III*. A subset of our models predict stable "chemical layering" (regime *II*) as either a preserved primordial layer (for high $B$) or a dense ROC layer (near-zero $B$ and low $\lambda_{\mathrm{prim}}$) in the lower mantle. For moderate $B$ and variable $\lambda_{\mathrm{prim}}$, primordial and ROC heterogeneity can coexist in a fully convecting mantle (regime *III*) with several distinct preservation styles. In contrast to previous work (Gülcher et al., 2020b), we do not observe any models with pervasive convective mixing of ROC as well as primordial mantle materials (regime *I*) in this study.

### 3.1.1 Chemical layering (regime II)

Models within regime *II* display strong compositional layering of the mantle after 4.5 Gyr of model evolution. These models span a wide range of compositional structures as well as tectonic styles across two subregimes (see Fig. 2):

**Primordial layer with topography *(II.T)*:** For large $B$ (>0.7), the initial layered configuration is preserved throughout model evolution. Both mantle down- and upwellings developing from the thermal boundary layers at the base of the litho-

sphere and the CMB, respectively, are deflected at the compositional interface. Subsequently, persistent double-layered convection develops with variable topography sustained at the interface as supported by flow in both layers. While primordial material largely remains confined to the lowermost mantle, thin tendrils of ROC are entrained into the lower layer and reach the lowermost mantle (Fig. 4a). Ultimately, primordial heterogeneity occupies about 60 v% of the lower mantle, in contrast to a mere 0.2 v% of ROC heterogeneity. The tectonic style is characterized by dominant mobile-lid behaviour ($M$>1) with short

intervals of low mobility ($M$<1, Fig. B1). This regime is similar to that described in Kellogg et al. (1999) and in Gülcher et al. (2020b).

**Post-overturn ROC layering *(II.R$_S$ and II.R$_E$)*:** For near-zero $B$ and $\lambda_{\mathrm{prim}} \leq 100$, models display (semi-)stable chemical stratification of the mantle with a stagnant-lid *(II.R$_S$)* to mobile-lid *(II.R$_E$)* tectonic style. In contrast to the previously described subregime, the chemical layering is formed after a large-scale overturn and is characterized by a thick layer of ROC at the base

of the mantle (>7 v%). Initially, double-layered convection is sustained for several 100s Myr as both weak lower-mantle upwellings and upper-mantle downwellings are deflected at the compositional interface in the mid mantle. However, progressive heating and cooling of the lower and upper mantles, respectively, promotes a whole-mantle overturn at $\approx$0.9 Gyr. Much of the primordial material in the lower mantle then reaches the upper mantle and is subsequently processed by extensive near-surface melting. Consequently, a large volume of basaltic crust forms that soon sinks to the lower mantle. The intrinsic high density of

this ROC (Fig. A2a) precludes any further entrainment by upwelling plumes. A small amount of primordial material (<4 v%) is preserved in the uppermost lower mantle, either as a thin diffuse primordial-material enhanced region (regime *II.R$_S$*, $\lambda_{\mathrm{prim}}$=10, Fig. 3b) or small coherent blobs (regime *II.R$_E$*, $\lambda_{\mathrm{prim}} \geq$30, Fig. 3c). Subsequent model evolution is characterized by chemical stratification with a ROC layer that mostly fully covers the CMB and small-scale convection within this layer, as it is heated

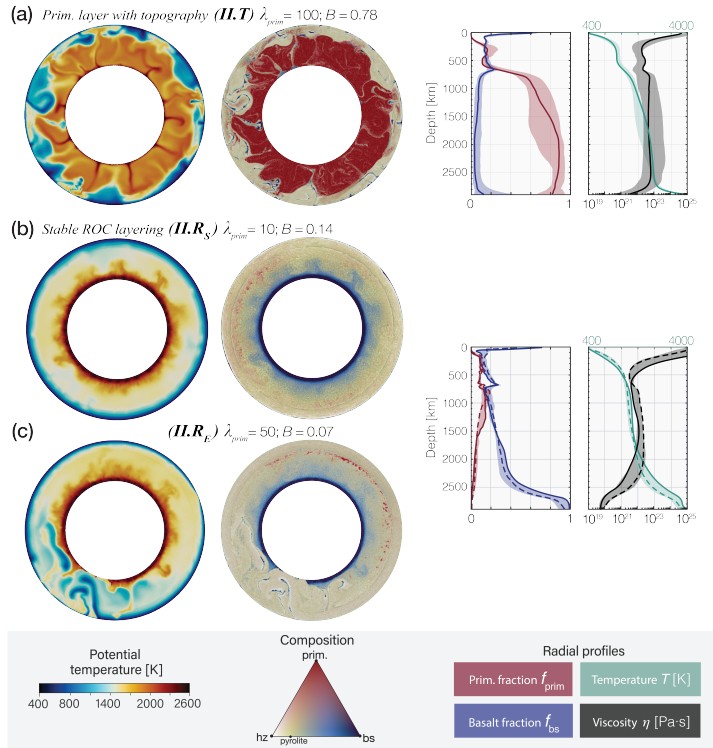

**Figure 3.** Left: Mantle sections of potential temperature (left) and composition (centre) for the two styles within regime I (chemical layering) at 4.5 Gyr model time. $\lambda_{prim}$ and $B$ as labelled. (right) corresponding profiles of primordial fraction, basaltic fraction, temperature and viscosity. These profiles are radially averaged and time-averaged (4.25-4.5 Gyr). The shaded region indicates the range for all models in any given subregime. Dashed lines refer to the case shown in panel (c).

from below. The upper mantle is strongly depleted (harzburgitic), preventing any significant further melting. In this stratified

mantle, convective vigour remains low. The radially averaged temperature and viscosity profiles (Figs. 3b-c) highlight the layering of the mantle, e.g., by showing an intermediate thermal boundary layer. For $\lambda_{prim} \geq 30$ (regime $II.R_E$), plates sporadically sink into the lower-mantle, as mobility $M$ fluctuates over time (Fig. B1c). For $\lambda_{prim} \leq 10$ (regime $II.R_S$), the tectonic style is characterized by a stagnant lid (low $P$ and $M$, Figs. B1b). Cases with $\lambda_{prim}=1$ display the same style of surface tectonics and of ROC layering as those with $\lambda_{prim}=10$ (see Table C1), but are not shown in Figure 3.

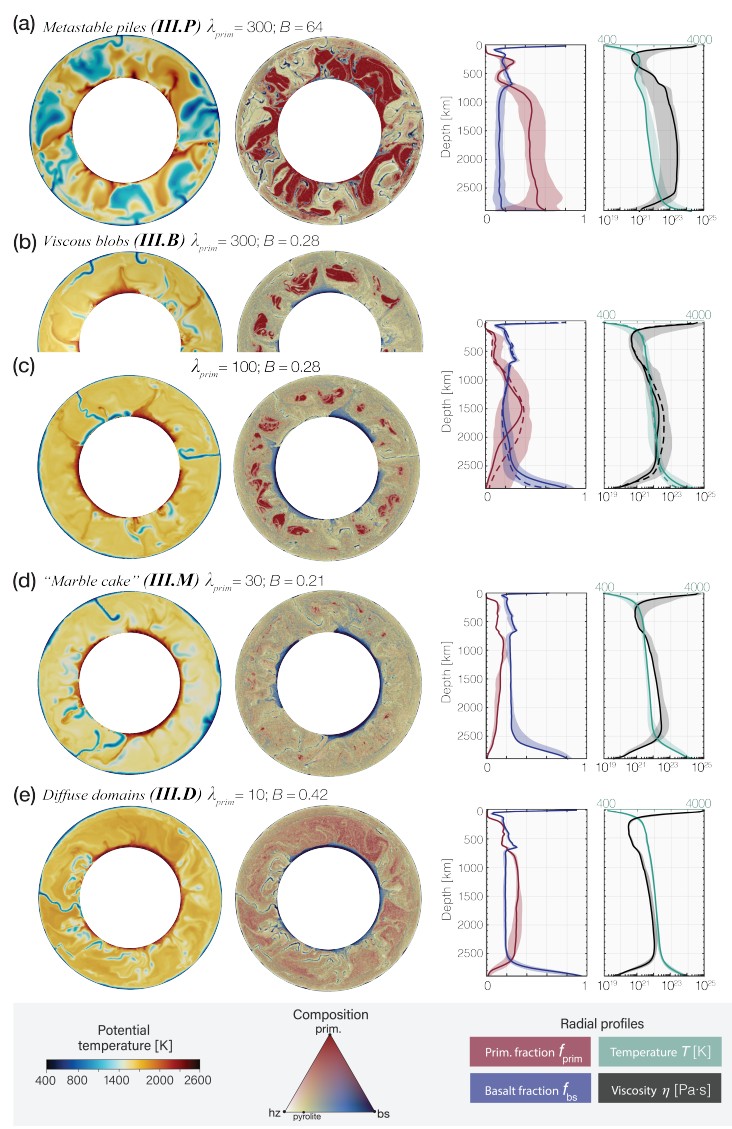

**Figure 4.** Mantle sections (left) and radial profiles (right) for all sub-styles in regime II (partial chemical heterogeneity) in the first model suite. Dashed lines refer to the case shown in panel (b). For detailed description, see Figure 3 caption.



### 3.1.2 Partial chemical heterogeneity (regime III)

The second regime is characterized by whole-mantle convection and moderate compositional heterogeneity preservation in the lower mantle after 4.5 Gyr of model evolution. This regime covers the parameter space of moderate-to-large viscosity contrasts ($\lambda_{\mathrm{prim}} \geq 10$) and low-to-moderate buoyancy ratios $B$ (see Fig. 2). All models display plate-like tectonic behaviour for most of their evolution (Figs. B1e-f). Their evolution starts with double-layered convection and subsequent heating/cooling of the lower/upper mantle, which eventually drives a global-scale overturn (after $\approx 1$ Gyr) with significant crustal formation and ROC subduction, such as in regimes *II.R*. During this overturn, the strong primordial material is separated into several blobs and/or streaks. These primordial domains are slowly eroded and entrained by the convecting pyrolitic mantle over billions of years. The final extent of heterogeneity survival depends on the physical parameters of the primordial material; three subregimes are established:

**Viscous blobs (*III.B*):** For a majority of this regime ($\approx 0.15 < B < 0.55$), primordial material is ultimately preserved as blobs in the mid mantle. Deformation due to mantle convection is mainly focused along narrow up- and downwelling conduits (Figs. 4b,c), and blobs are variably preserved in poorly-mixed regions in-between. These blobs periodically rotate and/or are displaced laterally as they are passed by sinking slabs. Occasionally, two blobs coagulate or are separated again, as the convection patterns re-organise through time. When slabs reach the CMB they push away basaltic piles to the side, disconnecting the otherwise continuous basaltic layer at the base of the mantle. The total volume of the preserved blobs strongly depends on the viscosity contrast $\lambda_{\mathrm{prim}}$. High $\lambda_{\mathrm{prim}}$ result in a delayed overturn and more-localised deformation, and hence slow entrainment of primordial material by the convecting mantle (Figs. 2, 4b). The final lower-mantle primordial ($\chi_{\mathrm{prim}}^{\mathrm{LM}}$) and ROC ($\chi_{\mathrm{ROC}}^{\mathrm{LM}}$) heterogeneity range from 8-40 v% and 1.5-6 v%, respectively. This style of material preservation is similar to that described in (Ballmer et al., 2017a; Gülcher et al., 2020b), and is here established with the presence of ROC (as piles atop the CMB and "marble cake" streaks throughout the mantle).

**Metastable primordial piles** (*regime III.P*)**:** For models with a moderate-to-high buoyancy ratio $B$ ($\approx 0.55$-0.7), primordial material is preserved as large primordial blobs and streaks that are mostly confined to the lowermost mantle. Initial model evolution is similar that in "viscous blobs" models (regime *III.B*), yet most primordial blobs ultimately settle near the CMB due to their relatively large negative buoyancy. Some of the blobs are occasionally pushed up from the CMB by convective currents and intermittently float through the mid-mantle. In contrast to other partial heterogeneity styles, short intervals with low mobility occur in this sub-regime (Fig. B1e). With the addition of the presence of dense ROC heterogeneity (1.5-4 v%) throughout the lower mantle as thin streaks, this regime is similar to previously described as "metastable piles" (regime *III-p*) in Gülcher et al. (2020b).

**"Marble cake" mantle (*III.M*):** A narrow subregime is detected for low $B$ and $\lambda_{\mathrm{prim}}$ in which the final mantle composition is characterized by the preservation of relatively much ROC (4.5-6.5 v%), and relatively little primordial heterogeneity (<5 v%). In addition to large ROC piles in the lowermost mantle, chemical heterogeneity takes the form of thin streaks of recycled (ROC) and primordial material in an otherwise well-mixed mantle. The small density contrasts and low-to-moderate viscosity





contrasts between primordial and ambient-mantle materials enhance mantle mixing. Ultimately, the mantle is mostly made up of a "marble cake" with both ROC and primordial streaks.

**"Diffuse" primordial domains** *(III.D)*: In this sub-regime, primordial heterogeneity is not preserved as discrete blobs or streaks (e.g., with sharp boundaries as in regime *III.B*) but is rather diffusely distributed throughout much of the lower mantle (Fig. 4e). It occurs for low $\lambda_{\mathrm{prim}}$ (=10) and moderate $B$ (0.35-0.50). The small viscosity contrast between primordial and ambient mantle material enhances mantle mixing and primordial material entrainment, such that discrete domains cannot be sustained in the convecting mantle. Nevertheless, significant amounts (2-7 v%) of primordial heterogeneity can survive in

the lower mantle for 4.5 Gyr model time. As in regime *III.B*, ROC is preserved as piles in the lowermost mantle, as well as streaks throughout the mantle. With these additional structures, regime *III.D* is similar to the diffuse-domain regime detected in Gülcher et al. (2020b).

### 3.2 Influence of initial layering

In all our cases, the final bulk composition of the ambient (non-primordial) mantle ($f_{\mathrm{bs,amb}}^{\mathrm{final}}$) depends on the initial volume of primordial material volume and the fraction of primordial material that is processed in the upper mantle (melting) (see eq. (3)). As the processed primordial material is converted to a 40:60 mechanical mixture of basalt and harzburgite (see Methods), the final amount of ROC heterogeneity in the ambient mantle also depends on the amount of primordial material that is processed through the upper mantle, and thus on the style of primordial-heterogeneity preservation. In the models discussed so far, $f_{\mathrm{bs,amb}}^{\mathrm{final}}$

attains highly variable values, depending on $\chi_{\mathrm{prim}}^{\mathrm{pres}}$. For cases with significant processing of primordial material through upper-mantle melting (regimes *II.R_{S,E}* and *III*), $f_{\mathrm{bs,amb}}^{\mathrm{final}}$ reaches very high values ($\approx 0.30$). These values are significantly higher than, for example, those of pyrolitic mantle composition ($f_{\mathrm{bs,amb}}^{\mathrm{final}} \approx 0.2$). Hence, many of the models in the first suite are not directly applicable to Earth.

Hereafter, we aim at evaluating models with a comparable final ambient-mantle bulk composition (and that is similar to that

of Earth). In order to achieve this, in model suite 2, we systematically vary the primordial layer thickness ($D_{\mathrm{prim}}$) as a function of parameters $B$ and $\lambda_{\mathrm{prim}}$. We choose a target value of the final ambient mantle composition $f_{\mathrm{bs,amb}}^{\mathrm{final}} = 0.25$, which is similar to the value suggested for Earth, particularly if considering that the ambient lower mantle may be moderately enriched in ROC compared to the upper mantle in the present-day (e.g., Murakami et al., 2012; Ballmer et al., 2015; Mashino et al., 2020; Yan et al., 2020). To reach this final target value, we consider the results of the first model suite and eq. (3) to estimate $D_{\mathrm{prim}}$ for a

given set of $B$ and $\lambda_{\mathrm{prim}}$ (for details, see Appendix A4).

#### 3.2.1 Primordial layer thickness

Final lower-mantle primordial and ROC heterogeneity ($\chi_{\mathrm{prim}}^{\mathrm{LM}}$ and $\chi_{\mathrm{ROC}}^{\mathrm{LM}}$) in models of the second suite are summarized in Fig. 2b and Table C2. In these models, the initial primordial layer thickness ranges from 1569 km to 1844 km (i.e., 60-75 v% of

the lower mantle). These initial conditions lead to final ambient-mantle bulk compositions of $f_{\mathrm{bs,amb}}^{\mathrm{final}} = 0.24$-$0.26$ (i.e., very

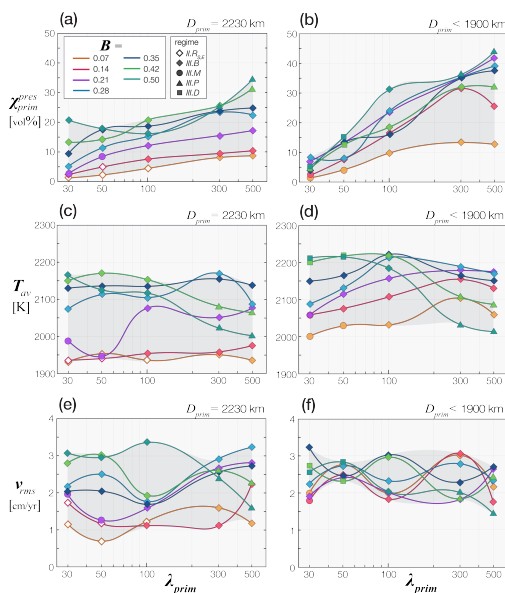

**Figure 5.** Output quantities time-averaged between 4.25 and 4.5 Gyr for model suite 1 (left) and model suite 2 (right). **a-b)** Primordial preservation factor $\chi_{\text{prim}}^{\text{pres}}$ in v% of initialised primordial material; **c-d)** average mantle temperature; **e-f)** convective vigour. The horizontal axis represents the primordial viscosity contrast $\lambda_{\text{prim}}$, while colours denote different $B$ used in the models.

close to our target value). The final amount of ROC heterogeneity in the lower mantle is systematically lower and (sub)regime boundaries are shifted compared to the first model suite (Fig. 2). The "partial chemical heterogeneity" regime (*III*) is expanded in the parameter space at the expense of regimes *II.R$_S$* and *II.R$_E$*, and the "marble cake" subregime (*III.M*) is shifted to even lower $B$. In all experiments in this model suite, the preserved primordial domains are slightly more diffuse compared to the first

model suite, which results in an expansion of the parameter space in which the "diffuse domains" subregime is stable (*III.D*). Most notably, the fraction of initialised primordial material that is preserved after 4.5 Gyr of mantle stirring ($\chi_{\text{prim}}^{\text{pres}}$) is similar in the second suite as in the first suite, and for $\lambda_{\text{prim}} \geq 100$ it is even higher (5a-b). Moreover, the final internal temperature of all experiments in the second model suite are systematically higher than those in model suite 1 ($T_{\text{av}}$, Fig. 5c-d), while the convective vigour in the experiments are more similar to one another ($v_{\text{rms}}$, Fig. 5e-f).

Figure 6 shows the temporal evolution of the radially averaged compositional profiles for selected experiments in both model suites. The compositional overturn (vertical black dashed line in Figure) occurs earlier for models with a thinner primordial layer, as convective instabilities more rapidly form in the upper layer to drive the overturn (see Table C2). With a primordial-dominated lower mantle (model suite 1), primordial material is immediately swept to upper-mantle depths during the overturn as being displaced by the slabs that sink into the lower mantle (see left panels of Fig. 6). Subsequent melting of primordial

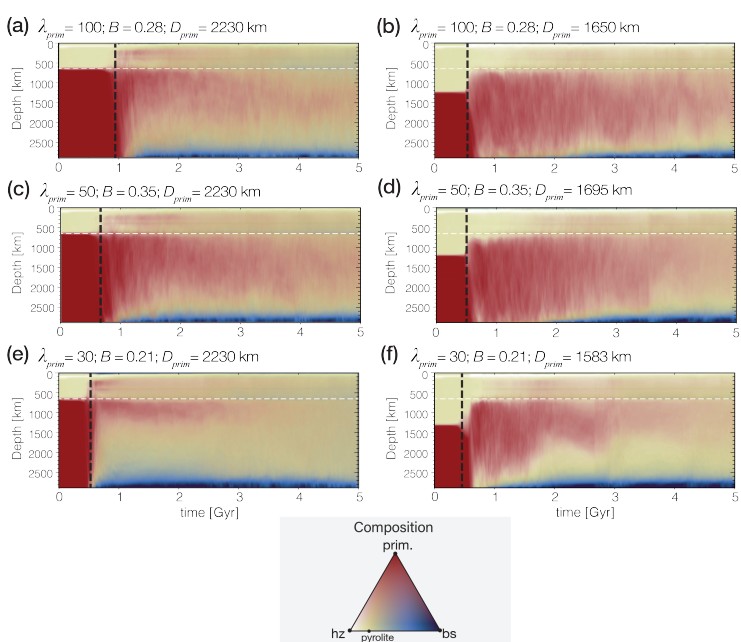

**Figure 6.** Temporal evolution of the radially-averaged compositional profiles of selected models. Models with fixed $D_{prim}$=2230 km (model suite 1) on the left, and their corresponding model with lowered $D_{prim}$ (from model suite 2) on the right for given $B$ and $\lambda_{prim}$: **(a-b)** $B$=0.28, $\lambda_{prim}$=100, **(c-d)** $B$=0.35, $\lambda_{prim}$=50, **(d-e)** $B$=0.21, $\lambda_{prim}$=30. The black dotted line indicates the onset of the compositional overturn. In model suite 1, but not in suite 2, significant amounts of primordial material immediately reach the upper mantle after the overturn.

material and crustal recycling leads to a quick buildup of basaltic piles in the lowermost mantle. For smaller primordial layer thickness as in model suite 2 (right panels of Fig. 6), the primordial layer detaches from the CMB during the overturn and is swept to mid-mantle depths without immediately reaching the upper mantle. At these mid-mantle depths, the primordial material remains relatively stable, and the formation of ROC piles is delayed compared to model suite 1. The detailed dynamics during the compositional overturn are shown in Fig. 7. Primordial domains are more fragmented for the case in model suite

1 (panel a) than in model suite 2 (panel b). Furthermore, the lower extent of melting in models with a thinner bridgmanitic layer (suite 2) explains the higher internal temperatures (Fig. 5d). As found in Lourenço et al. (2016); Nakagawa and Tackley (2012), melting is an efficient mechanism to remove heat from the mantle.

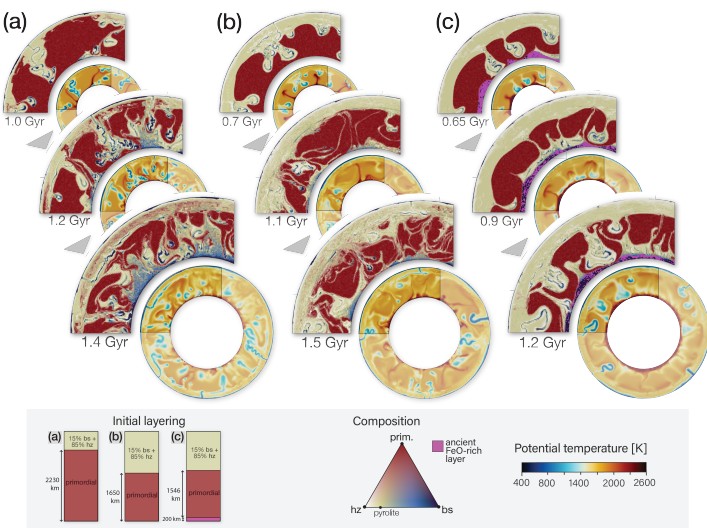

**Figure 7.** Snapshots during the the compositional overturn phase for three models with equal $B$ (0.28) and $\lambda_{prim}$ (= 100), but a different initial compositional layering (as shown on the bottom left). The panels provide the potential temperature field, and a zoom-in on the compositional field.

### 3.2.2 Additional basal FeO-rich layer

We further explore several additional cases that include an additional FeO-rich layer just above the CMB as an initial condition
(see Appendix A1). The results of these cases are summarised in Figure 2b and Table C2. The imposed material properties of the FeO-rich layer are the same as for basalt (i.e. intrinsically dense, see Fig. A2). Notably, the ancient dense basal layer aids the preservation of bridgmanitic primordial material over time (higher . During the compositional overturn, the FeO-rich layer (pink colour in Fig. 7c) remains at the base of the mantle, while the bridgmanitic material is swept to mid-mantle depths. Significant amounts of heat can be retained in this negatively buoyant dense layer that acts an insulating boundary layer above
the core. Plumes readily form on top of this layer and are less hot than those directly forming above the CMB in model suites 1 and 2, resulting in a less vigorous compositional overturn and thus an increased coherency of primordial domains. Downgoing slabs separate the initially laterally-connected dense layer into distinct piles, and they also transport ROC to the lowermost mantle, which is then added to the piles of ancient FeO-rich material (see last two panels of Fig. 7c). The incoming ROC material is cooler and thus denser than the hot ancient FeO-rich material. Accordingly, it sinks through the ancient layer to settle
directly above the CMB. Over time, much of the ancient FeO-rich material is progressively entrained by mantle upwellings and ultimately processed by near-surface melting. Only a small fraction is preserved after 4.5 Gyr (Table C2), preferably located in the uppermost parts of some thermochemical piles (Fig. 8) and underlain by ROC. The general trend in these piles is aging-





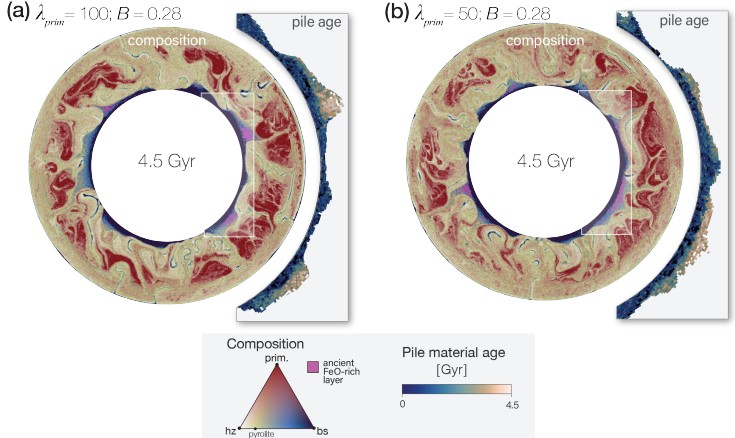

**Figure 8.** Final snapshots (at 4.5 Gyr) of composition for selected models, $\lambda_{prim}$ and $B$ as stated in the panels. The models are initialised with a 200 km-thick FeO-rich basal layer (pink), overlain by a (a) 1546 and (b) 1514 km-thick primordial layer. The panels also show a zoom-in on the formation age of ROC within the piles (defined as $f_{bs} > 0.6$ and $T_{pot} > 2000$ K). The formation age quantifies the time since the last melting episode.

upwards: the youngest, most recently subducted (cool) ROC material is at the bottom with progressively older ROC material upwards, and the ancient (hot) FeO-rich material at the very top. Thereby,the piles are thermally stratified, and internal small-

scale convection occurs in the lowermost, dense layer. Ultimately, the coexistence of large, coherent bridgmanitic primordial domains with basaltic piles (ROC and ancient) is robustly predicted, even for low $\lambda_{prim}$ (see Table C2).

### 3.3    Relating dense piles, viscous blobs and mantle dynamics

Finally, we analysed the potential relationship between dense piles, viscous blobs and mantle dynamics for a model that displays both piles and blobs (model $M_{300D}$, regime *III.B*, $\lambda_{prim}$=300 and $B = 0.28$, illustrated in Fig. 4b). Figure 9 shows 2D

histograms relating selected model output quantities with one another. These quantities are pile characteristics (pile height; formation age of pile material) and mid-mantle characteristics (primordial material fraction, radial velocity, and temperature anomaly). The formation age of pile material is volume-averaged over the full pile thickness and weighted by the basaltic fraction. Mid-mantle quantities are volume-averaged over a depth range of 1000-2000 km. All quantities are evaluated for each mantle column (512 columns in the model domain) and for six different timesteps (3072 columns total). Each row in

each histogram is scaled by the total counts of that row. Several relations between these parameters can be inferred: first, the presence of primordial material in the mid-mantle is mostly related with little radial dynamics (near-zero $v_z$, Fig. 9a). Similarly, most piles are stable at near-zero radial velocity, although thick (>100 km) piles are also stabilized in regions of upwellings. Notably, the absence of piles ($D_{prim}$=0, bottom row in Fig. 9e) relates to columns where downwellings occur ($v_z$ <0). Second,

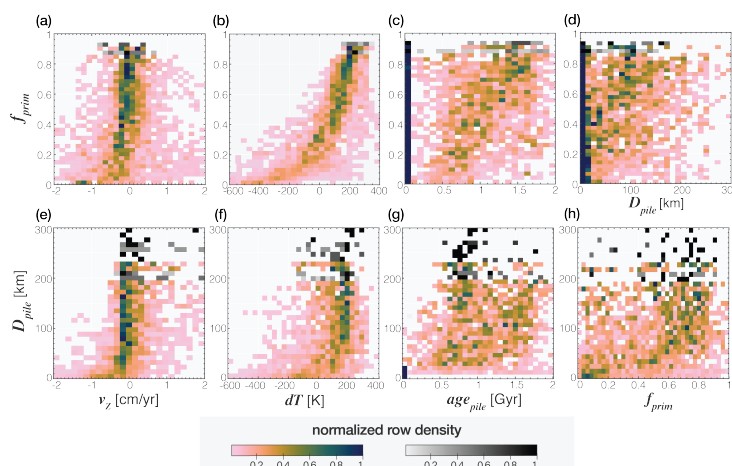

**Figure 9.** Each panel shows a 2D histogram that relates several selected output quantities of a representative model displaying both primordial blobs and dense piles underneath (regime *III.B*, model $M_{300D}$, Fig. 4b). For six timesteps (4.25 - 4.5 Gyr with steps of 0.05 Gyr), selected quantities were calculated for each radial column in the model: pile height $D_{pile}$, formation age of pile material $age_{pile}$, mid-mantle primordial material fraction $f_{prim}$, mid-mantle radial velocity $v_z$, and mid-mantle temperature anomaly $dT$. For the thermochemical pile, thresholds are $f_{bs} > 0.6$ and $T_{pot} > 2000$ K. The mid-mantle is defined as the depth range of 1000-2000 km, and quantities are volume-averaged in this depth-range. Each row in the histograms is normalized by the total counts of the row. Rows are plotted in greyscale if the number of counts in this row represents less than 1 % of the total counts.

the presence of much primordial material in the mid-mantle is associated with anomalously high temperatures (Fig. 9b). This
indicates that large blobs (high $f_{prim}$) are typically slightly warmer than average (also see Figs. 4b,c). There is also a positive
correlation between pile height and mid-mantle temperature anomaly, although less evident. Whereas the absence of piles is
associated with cool downwellings (bottom row in Fig. 9f), large piles are often related to overlying warm primordial blobs
(see below). Although the histograms remain scattered, there is a slight positive trend in formation age of pile material versus
mid-mantle primordial fraction, and formation age versus pile height (Figs. 9c,g). This indicates that piles containing older
material are more often overlain by primordial domains than younger piles, and old ROC tends to be preserved near to top of
the piles (see above). Finally, figures 9d,h indicate a spatial relationship (in radial direction) between the presence of primordial
material in the mid-mantle and the height of the underlying ROC material. Even though this relationship is not strict, it indicates
that the highest ROC piles tend to be centered just below the largest primordial blobs. This configuration away from up- and
downwellings (Fig. 9b,f) appears to promote the coupled preservation of both these thermochemical domains.



## 4 Discussion

### 4.1 Styles of mantle mixing in terrestrial planets

Many of the identified dynamic styles of primordial material preservation in our prior study (Gülcher et al., 2020b) are here established in the presence of plate-like behaviour and recycling of oceanic crust into the deep mantle. This indicates that primordial material preservation robustly occurs for significant rheological contrasts, regardless of tectonic behaviour, and also in the presence of dense, recycled materials in the lower mantle. In this study, primordial material preservation is practically independent of primordial-layer thickness, and is advanced for an additional thin layer of FeO-rich that is initially present just above the CMB. In contrast to (Gülcher et al., 2020b), we did not detect any model in which both primordial and recycled materials are efficiently mixed throughout the mantle (regime $I$ in Gülcher et al. 2020b). We expect such a behavior to occur for small rheological contrasts of primordial material (low $B$ and $\lambda_{prim}$) in combination with a lower density anomalies of ROC than modelled here (see Fig. A2) (Nakagawa and Tackley, 2014). In fact, the density and rheological properties of ROC in the lowermost mantle are still under debate (e.g., Ono et al., 2005; Hirose et al., 2005; Xu et al., 2008; Nakagawa et al., 2010; Trønnes et al., 2019). These properties may affect the segregation, accumulation and survival of ROC in the deep mantle (e.g., Christensen and Hofmann, 1994; Nakagawa et al., 2010; Bower et al., 2013; Li et al., 2014b; Mulyukova et al., 2015), and hence the mantle heterogeneity regimes detected in this study.

The relevance of each regime for terrestrial planet evolution depends on the actual initial compositional profile of the mantle at the onset of long-term solid-state convection. The initial conditions of most of our models are mainly motivated by an enrichment of a thick, basal layer in $(Mg,Fe)SiO_3$ bridgmanite (and hence in silica with respect to the overlying mantle). This may occur due to fractionation during magma-ocean or basal magma-ocean (BMO) crystallization (e.g., Labrosse et al., 2007). Indeed, bridgmanite is the liquidus phase in a magma ocean over a wide compositional and pressure range (e.g., Ito and Takahashi, 1987; Caracas et al., 2019). Alternatively, incomplete core-mantle equilibration during the last stages of planetary accretion may have aided silica-enrichment of the lower mantle (Deng et al., 2019; Kaminski and Javoy, 2013). Moreover, chemical transfer between the outer core and the (basal) magma ocean in a hot early Earth may have caused delivery of $SiO_2$ to the BMO and extraction of iron from the BMO into the core, stabilising bridgmanite as the primary liquidus phase during prolonged periods of BMO crystallisation (e.g., Trønnes et al., 2019). Indeed, various studies have proposed $SiO_2$-exsolution from the core during secular cooling of the Earth or Earth-like planets (e.g., Badro et al., 2016; Hirose et al., 2017; Helffrich et al., 2018a; Rizo et al., 2019).

A magma ocean is also thought to have existed on Mars, and its solidification may have compositionally stratified the Martian mantle (Elkins-Tanton et al., 2003). Isotopic signatures of Martian meteorites indicate that convection did not fully homogenize the early Martian mantle (e.g., Jagoutz, 1991; Lee and Halliday, 1997; Foley et al., 2005; Marchi et al., 2020), consistent with interior models of planetary evolution (e.g., Reese et al., 2002; Zhang and O'Neill, 2016). For Venus, high temperatures and slow mantle cooling compared to Earth (in the absence of plate tectonics) naturally extend the predicted lifetime of a possible (basal) magma ocean (O'Rourke, 2020). Such a prolonged lifetime of a BMO likely caused a delay in terms of mixing of



any primordial stratification. In addition, due to the smaller tectonic activity of the lithosphere and smaller mantle Rayleigh number, primordial heterogeneity may be better preserved on Venus than on Earth.

**4.2 Observations of chemical heterogeneity in the Earth**

For planet Earth, any favorable mantle heterogeneity scenario should be consistent with several geophysical and geochemical constraints. First, any double-layering of the mantle can be ruled out by seismic imaging of recently (<200 Myrs) subducted lithosphere in the lowermost mantle (e.g., van der Hilst et al., 1997), and of deep-rooted plumes that rise through the entire mantle (French and Romanowicz, 2015). These observations are interpreted in terms of thermochemical convective flow that

encompasses the whole mantle (e.g., van Keken and Ballentine, 1998). Second, while their origin and nature remains debated, the two large low shear-velocity provinces (LLSVP) in the lowermost mantle are consistently observed in seismic tomography models (e.g., Li and Romanowicz, 1996; Ritsema et al., 1999; Trampert et al., 2004; Houser et al., 2008; Ritsema et al., 2011; French and Romanowicz, 2015). The origin of LLSVPs (e.g. purely thermal or thermochemical) remains debated upon. While a purely thermal nature of LLSVP has been suggested (e.g., Schubert et al., 2004; Schuberth et al., 2009; Davies et al., 2012),

it cannot account for the anticorrelation between shear-wave and bulk-sound velocity in the lowermost 500 km of the mantle (e.g., Su and Dziewonski, 1997; Antolik et al., 2003), unless post-perovskite can be stabilized in hot regions (Koelemeijer et al., 2016). Moreover, seismic waveform studies infer sharp velocity contrasts along their margins (e.g., Ritsema et al., 1997; Ni et al., 2002; Wang and Wen, 2004; To et al., 2005; Zhao et al., 2015), which is a strong indication for LLSVPs being compositionally distinct from the surrounding mantle. The sub-horizontal orientation of these margins supports pile-

like shapes of LLSVPs, i.e., further indicative of a compositional distinction, whereas purely thermal models require plume bundles with sub-vertical structures. A hybrid model with thermochemical plume bundles has been recently proposed (Davaille and Romanowicz, 2020), but appears to be also inconsistent with these orientations. Third, the preservation of ancient mantle heterogeneity through Earth's history is evident in the geochemical record. Early studies on isotopic signatures of oceanic island basalts support the the survival of old recycled oceanic crust (several Gyrs) in the deep mantle to present-day (e.g.,

Chauvel et al., 1992; Hofmann, 1997; Cabral et al., 2013). Moreover, recent studies of anomalies in the daughter nuclides of short-lived isotopic decay systems $^{182}$W/$^{184}$W and $^{142}$Nd/$^{143}$Nd in igneous rocks provide strong evidence for the persistence of primordial mantle heterogeneity that was formed within the first ∼50 Myrs of solar system history (e.g., Touboul et al., 2012; Rizo et al., 2016; Mundl et al., 2017; Peters et al., 2018). Further to this, high $^{3}$He/$^{4}$He signatures in ocean island basalts are indicative of an undegassed primordial source (Jackson et al., 2010). Finally, on Earth, plate tectonics is thought to have

operated for at least several Gyrs before present-day, with the exact timing of the onset of modern-style plate tectonic to be heavily arguable (Korenaga, 2013, and references therein).

These geochemical and geophysical constraints are best explained by geodynamic regimes that predict the coupled preservation of primordial material and ROC. These include the regimes *III-D* (diffuse primordial domains), *III-M* ("marble cake" mantle), and *III.B* (viscous blobs). Out of these styles, the primordial blobs style (*II-B*) is most commonly predicted, while the

diffuse domains and marble cake regime instead only occupy a narrow parameter window (Fig. 2). Model predictions for sub-regime *III-B* are also consistent with the stagnation of some slabs in a depth range that is similar to primordial-domain roofs,



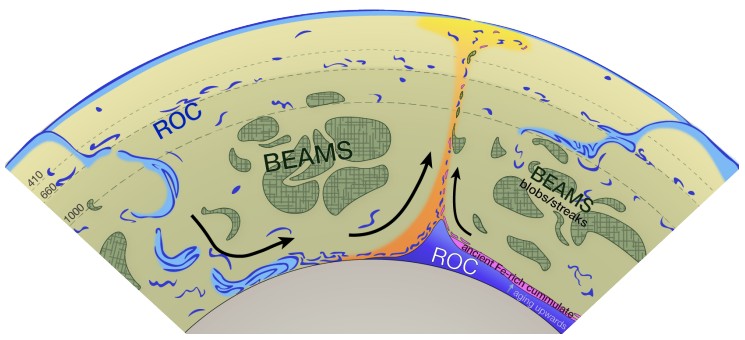

**Figure 10.** sketch of our proposed style of present-day mantle heterogeneity in the Earth. Recycled crustal material (ROC, dark blue) is present throughout the mantle as "marble cake streaks", with enhancement in the mantle transition zone (between 410 and 660 km depth). Moreover, the dense ROC material settles at the core-mantle boundary to form thermochemical piles. Any ancient basal layer may be preserved within these piles, here proposed on the roof of the piles as predicted by our models (see Figure 8). Finally, primordial $(Mg,Fe)SiO_3$-enriched material is preserved as blobs and streaks that reside in the mid-mantle (dark green) due to its intrinsic strength.

while other slabs readily sink into the deep mantle(Fukao and Obayashi, 2013), as seen in Figs. 4, 8 and Suppl. Videos, (Gülcher et al., 2020a)). Moreover, the presence of viscous domains in the mid-mantle may explain the observation of widespread sharp seismic velocity contrasts in the uppermost lower mantle (850-1100 km depth), usually occurring away from major upwellings

and downwellings (Jenkins et al., 2017; Waszek et al., 2018). In terms of seismic tomography, the lack of clear evidence for primordial domains in the mid-mantle may be related to a trade-off between the seismic effects of their compositional (intrinsically fast bridgmanite, Wentzcovitch et al., 2004) and thermal (slightly warmer than the ambient mantle, see Fig. 9c) anomalies.

### 4.3 A recipe for Earth's lower mantle: "marble cake" plus "plum pudding"

Based on our results, we propose a hybrid style of present-day Earth mantle heterogeneity, which is illustrated in Fig. (10). This style includes the coexistence of viscous primordial material and dense ROC material in a convecting mantle. Primordial domains are present as streaks and/or blobs that reside in the mid-mantle. Several of these streaks and/or blobs form a larger composite bridgmanitic structure (100s-1000 km) as convective flows are organised around them. Except for very large $\lambda_{prim}$, this prediction is somewhat different from that in (Ballmer et al., 2017a), who predict large coherent blobs. Such composite

bridgmanitic structures are consistent with a recent study on the (in)visibility of the iron-spin transition in global tomography models (Shephard et al., 2020). ROC material is present as "marble-cake" streaks throughout the mantle with local enhancement in the mantle transition zone (MTZ) and at the CMB (i.e., forming thermochemical piles) (as in Nakagawa and Buffett, 2005; Yan et al., 2020). Piles in this regime are mostly of recycled origin, but may include ancient FeO-enriched material that is





preserved near their top, if a FeO-rich basal layer ever existed in the early Earth. In this scenario for mantle composition and

structure, the upper mantle is on average pyrolitic, while the lower mantle is significantly more enriched in silica, in line with previous studies (e.g., Murakami et al., 2012; Ballmer et al., 2015; Mashino et al., 2020; Yan et al., 2020).

### 4.4 Thermochemical pile layering

Thermochemical piles formed in our models comprise of ROC with varying formation ages. For models including an ancient FeO-rich basal layer, piles consist of both recycled and ancient materials (e.g. a "basal melange", Tackley 2012). Towards the

roof of the piles, the formation age of ROC increases, while the overall ROC content decreases (Fig. 8). A radial distinction into regions of a high- and low-ROC content within piles, was also found by Mulyukova et al. (2015). For a high buoyancy ratio of basaltic material, the thermochemical piles consist of a high-density basal-layer covering nearly the entire CMB, overlain by high-topography piles with a much lower fraction of ROC (Mulyukova et al., 2015), as is the case for the thermochemical piles in our models. Moreover, in our experiments, ancient FeO-rich material survives for several Gyrs at the roof of some

piles (Fig. 8). Compositional layering of ancient vs. recycled domains within LLSVPs has been previously proposed (Ballmer et al., 2016; Trønnes et al., 2019), yet with an inverted maturity gradient as found here.

Yet, chemical layering strongly depends on the density profile and rheological properties of the material that makes up the thermochemical piles, which remain poorly constrained. We modelled the ancient FeO-rich basal layer to have the same physical properties as ROC. Final preservation of an ancient basal layer would be advanced, and the maturity gradient potentially

inverted, if the layer were enriched in FeO or $SiO_2$ relative to ROC. Such an enrichment may or may not occur depending on the formation scenario of the ancient basal layer, e.g., the style of magma-ocean crystallization (e.g., Labrosse et al., 2007; Ballmer et al., 2017b), or core-mantle interaction (e.g., Hirose et al., 2017; Trønnes et al., 2019). In any case, ancient crustal rocks may have been enriched in MgO and hence less dense than (relatively) young basalt (Herzberg and Rudnick, 2012), supporting the layering sequence within piles as predicted by our models.

### 435 4.5 Relationship between piles, blobs and mantle dynamics

Our results show that there is a slight positive correlation between pile height and overlying primordial material (Fig. 9). Piles and blobs both mostly reside well away from lower-mantle downwellings, as they are pushed away from them (Suppl. Videos, Gülcher et al. (2020a)). Thermochemical piles that are overlain by large primordial blobs are on average comprised of older material than those that are not overlain by blobs (Fig. Fig. 9d,h). This result suggests that primordial domains can shield piles

from any incoming subducted material, and aid their longevity.

These results further contribute to the ongoing debate about whether thermochemical piles are intrinsically stable features in the deep mantle which spatially determine mantle convective patterns (Dziewonski et al., 2010; Torsvik et al., 2014), or are rather pushed around by subduction zones (McNamara and Zhong, 2004). We observe that downgoing slabs are responsible for the spatial distribution of piles (as well as viscous blobs), as the piles are laterally moved by the stiff downwellings (subduction

zones). This control of subduction zones on the spatial distribution of piles (and not the other way around) has been noted in previous studies (e.g., McNamara and Zhong, 2004; Schierjott et al., 2020).



### 4.6 Shift in mantle dynamics and chemical sampling of the deep Earth

An interesting prediction of our models involves the rapid change of geodynamic style when the compositional layering breaks down ("overturn"), marking the beginning of whole-mantle convection (see Figs. 7, B1.). From the overturn onwards, plate-tectonic behaviour ($P$ and $M$) robustly occurs in the models (i.e., within the relevant regime III). The timing of the overturn in these (sub)regimes varies between 0.5 and 2.5 $\mathrm{Gyr}$ of model evolution, mainly depending on the intrinsic strength of the primordial layer. For Earth, a major shift in geodynamic regime is commonly proposed to mark the onset of 'modern-style' plate tectonics (e.g., Bédard, 2018; Condie, 2018, and references therein), shifting from either an (episodic) stagnant lid (e.g., O'Neill et al., 2016; Bédard, 2018; Lenardic, 2018) or a sluggish lid (Rozel et al., 2017; Foley, 2018; Lenardic, 2018; Lourenço et al., 2020) with mostly vertical tectonics to a mobile-lid style with horizontal "plate-like" tectonics. Estimates for the timing of this major shift vary between studies (e.g., 2 - 3.2 Ga, Korenaga 2013, and references therein). According to our model predictions, the breakdown of ancient compositional layering may have triggered sich a change in surface-tectonic style. The related stirring of the mantle during the overturn may have also caused a drastic change in upper-mantle composition, likely followed by the oxygenation of the atmosphere, with implications for the evolution of life (Andrault et al., 2017). Model behaviour after the onset of plate tectonics is furthermore marked by variable mobility, indicative of episodes of vast tectonic activity and material recycling.

Another implication of this geodynamic shift is that deep-rooted mantle plumes can reach the upper mantle for the first time after the overturn. Accordingly, the geochemical signatures of mafic rocks of this age should carry stronger lower-mantle signatures compared to older mafic rocks. Many geochemical studies show evidence for a rapid, widespread change in isotopic signatures in basalts around $\approx$3 Ga (e.g., Gamal El Dien et al., 2020, and references therein). For example, $\mu^{182}$W anomalies in basalts show a steady trend of mostly positive values for rocks of before $\approx$3 Ga, but display negative values in the modern mantle. This change has previously been attributed to inner-core segregation (Rizo et al., 2019) or the onset of deep slab subduction, recycling of crustal material and sediments (e.g., Liu et al., 2016; Rizo et al., 2019). Here we propose it may (additionally) be related to the onset of primordial-material entrainment by mantle plumes via whole-mantle convection. The primordial material could have obtained a negative $\mu^{182}$W signal through $SiO_2$-exsolution from the core (which has a strongly negative $\mu^{182}$ signature (e.g., Kleine et al., 2009) to the (basal) magma ocean during rapid initial cooling of the planet (e.g., Helffrich et al., 2018b; Trønnes et al., 2019; Rizo et al., 2019).

### 4.7 Model limitations and outlook

The work presented here is primarily a geodynamic study that establishes various feasible regimes of mantle dynamics and heterogeneity mixing/preservation as a function of initial conditions and primordial-material properties. In the future, a thorough integration with inter-disciplinary constraints on mantle heterogeneity is needed to further establish the applicability of some of these regimes for the Earth's (lower) mantle. For example, model results should be quantitatively compared with seismic constraints: piles are predicted to be anomalously hot and dense, while viscous blobs in our models are usually only slightly warmer than average mid-mantle temperatures.



Moreover, our numerical models remain a simplified approximation of nature. For example, our model resolution only allows us to detect relatively large heterogeneities (on scales of several km, or larger). Also, we do not include any felsic material in our models, which may contribute to lower mantle heterogeneity by recycling of continental material (Hofmann, 1997; Kawai et al., 2009; Stracke, 2012). Further, our models are limited in terms of geometry (2D spherical annulus). Indeed, it has been shown that mantle mixing is more efficient in 3D geometry, in which toroidal and poloidal flow components interact with each

other, than in 2D geometry for purely thermal models (Ferrachat and Ricard, 1998; Coltice and Schmalzl, 2006). However, it is not obvious that this prediction holds true for models with (strongly) composition-dependent rheology, since mantle flow may be even more efficiently guided around viscous blobs in 3D than in 2D geometry (Merveilleux du Vignaux and Fleitout, 2001).

Finally, internal (radioactive) heating is switched off in the current models. While significant heat production within primordial domains is likely to impede preservation, we note that bridgmanitic magma-ocean cumulates are unlikely to incorporate

any significant levels of highly-incompatible elements, including radioactive nuclides (Corgne et al., 2005).

## 5    Conclusions

We performed a numerical study to investigate the coexistence of several types of chemical heterogeneity with distinct physical properties in the lower mantle. Our results demonstrate that:

- Primordial and recycled heterogeneity may coexist in various styles in the mantles of Earth-like planets, depending on
495        their rheological properties (intrinsic density and viscosity).

- The final volume of primordial heterogeneity preserved in the mantle only very weakly depends on the initial thickness of the primordial layer in the range of values explored here (60-100 v% of the total lower-mantle).

- The addition of an ancient FeO-rich basal layer in the lowermost mantle promotes the preservation of coherent viscous blobs in the mid-mantle.

- The coexistence of viscous, primordial blobs in the mid-mantle with dense recycled (and partially ancient) piles in the lowermost mantle is a robust model prediction over a wide range of parameters.

- The presence of large viscous blobs in the mid-mantle stabilises underlying dense thermochemical piles, contributing to their preservation and longevity.

Our results provide a quantitative and testable framework for the coupled evolution of recycled and primordial materials in a

convecting mantle. For planet Earth, we suggest that the lower mantle may be in a hybrid state between "marble cake" and "plum pudding" style of chemical heterogeneity. Such a hybrid style of heterogeneity can reconcile geochemical evidence for ancient-rock preservation in the convecting mantle through the present day with seismic evidence for whole-mantle convection. Finally, the breakdown of a stratified system with double-layered convection towards a whole-mantle convective system with plate-tectonic behavior as predicted by our models may be relevant for a major geodynamic "shift" during the Archaean, as is

indicated by the geological and geochemical record.





*Code and data availability.* The numerical code is available for collaborative studies by request to P.J. Tackley. The data corresponding to the numerical experiments of this manuscript is too large to be placed online, but they can be requested from the corresponding author.

*Video supplement.* Video supplements are available at Zenodo under the identifier https://zenodo.org/record/4298777 (Gülcher et al., 2020a).





## Appendix A:  Mantle composition

### A1    initial layering set-up

The various initial conditions of chemical layering used for this study are illustrated in Figure A1. For the first model suite (panel a), a thick, primordial layer expands from the CMB to the top of the lower mantle (660 km depth). For cases in model suite 2 (panel b), this primordial layer thickness is decreased according the results from model suite 1 (see Appendix A4). Finally, selected cases are initialised as a three-layered system (panel c), in which a 200 km-thick FeO-rich layer is imposed

just above the CMB. The imposed material properties of the FeO-rich layer are the same as for basalt (i.e. intrinsically dense, see Fig. A2).

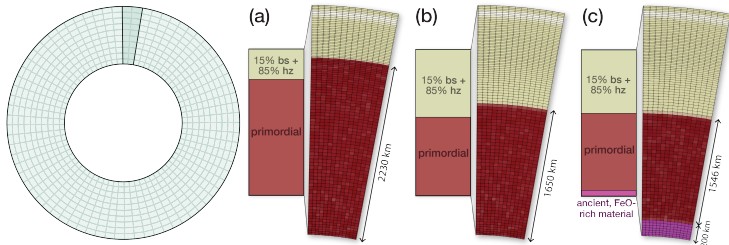

**Figure A1.** Schematic sketch and actual model domain showing the various chemical layering set-ups used for different model suites. The modelled domain is discretised in 512·95 cells, and resolved by over 1.2 million markers. Reference model ($\lambda_{\text{prim}} = 100$; $B = 0.28$) for **a)** Model suite 1; **b)** Model suite 2; **c)** 3-Layered set-up. Here, the primordial layer is moved away from the CMB and its thickness $D_{\text{prim}}$ is based on initial primordial material volume-preservation with respect to the case in model suite 2.

### A2    Density profiles of modelled materials

The density profiles of the relevant mantle materials in this study are plotted in Fig. A2. The density profiles of harzburgite and basaltic materials are consistent with those from Xu et al. 2008. The profile of primordial material is parametrised to be

consistent with a rocky material with a (Mg+Fe)/Si ratio of $\approx 1.0$ (such as bridgmanite). It resembles that of a solid solution of 40% basalt and 60% harzburgite in Xu et al. 2008 in the upper mantle. Linearly fitting this density profile to those obtained by experimental studies of pure $MgSiO_3$ bridgmanite (Tange et al., 2012) and $Mg_{0.87}Fe_{0.13}SiO_{0.3}$ bridgmanite (Wolf et al., 2015) materials at mid-mantle depths (1500 km), we estimate that our reference primordial material can be interpreted as corresponding to $Mg_{0.88}Fe_{0.12}SiO_2$, or any other material with a similar density profile. To explore the effects of buoyancy

ratio $B$ in both model suites, the density profile of primordial material is shifted throughout the mantle between $-18$ kg/m$^3$ and $+42$ kg/m$^3$ in steps of 6 kg/m$^3$. Thereby, $B$ in the models ranges between 0.07 to 0.78, corresponding to primordial material Mg# ranging from 0.9∼0.85.

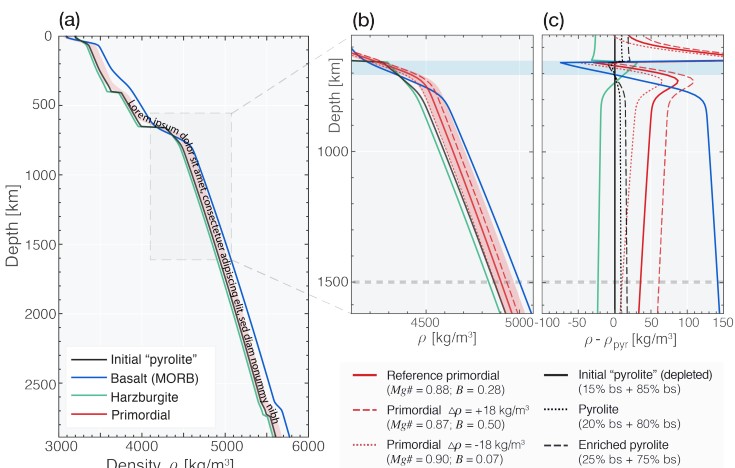

**Figure A2.** Density profiles for mantle materials used in our simulations. **a)** Density profiles for basalt (light blue), harzburgite (black), initial "pyrolite" (dark blue), and primordial material (red). **b)** Zoom-in on mid-mantle depths. Blue shaded area is the depth range where basaltic material is less dense than pyrolite (transition zone). The grey dotted line is at $1500$ km depth, for which the buoyancy number $B$ is calculated (see text). **c)** Relative density contrasts with depth for mantle materials relative to that of the initialized "pyrolitic" material. The red solid line represents our reference primordial material. The dashed and dotted red lines represent primordial material with a buoyancy shift $\delta B = +/- 0.21$. The initial "pyrolite" material is plotted as the black solid line, which is lightly depleted to that of present-day pyrolite ($20\%$bs and $80\%$hz, black dotted line). A more enriched pyrolite composition ($25\%$bs and $75\%$hz) is plotted as a black dashed line.

### A3  Visualisation and heterogeneity detection

While the composition of an individual tracer is either primordial or a projection on a 1D axis between basalt and harzburgite,

that of a grid cell can contain all three possible end-members (bs, hz, and primordial, with $f_{bs} + f_{hz} + f_{prim} = 1$). The cell composition is therefore visualised with a two-dimensional triangular colour map, given in Fig. A3a).

Two types of lower-mantle chemical heterogeneity are detected in the experiments: primordial ($\chi_{prim}^{LM}$) and recycled oceanic crust ($\chi_{ROC}^{LM}$). We define $\chi_{prim}^{LM}$ and $\chi_{ROC}^{LM}$ as the relative volumes of the lower mantle with fractions of primordial material of $f_{prim} > 0.6$ and of ROC of $f_{bs} > 0.6$,respectively (see Fig. A3b). The results reported in Figure 2 and Tables C1 and C2 are

time-averaged between 4.25 and 4.5 Gyr of model evolution.

### A4  Fitting ambient mantle composition

In order to compute $D_{prim}$ for the experiments in model suite 2, we rewrite eq. (3) within the following assumptions. First, we assume that the final volume of primordial material mixing depends primarily on the preservation factor of primordial material





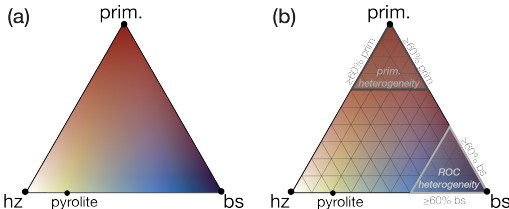

**Figure A3. a)** colour scale used in this study to show composition on a grid level, that can consist of harzburgite, basalt and primordial material; **b)** visualisation of the chemical heterogeneity detection. Primordial heterogeneity $\chi_{\text{prim}}^{\text{LM}}$ is defined as: $f_{\text{prim}} > 0.6$; ROC heterogeneity $\chi_{\text{ROC}}^{\text{LM}}$ is defined as: $f_{\text{bs}} > 0.6$.

and initial volume of primordial material:

$$V_{\text{prim,molten}}^{\text{final}} = V_{\text{prim}}^{\text{ini}} - V_{\text{prim}}^{\text{final}} = \left(1 - \chi_{\text{prim}}^{\text{pres}}\right) \cdot V_{\text{prim}}^{\text{ini}} \tag{A1}$$

with preserved primordial volume fraction $\chi_{\text{prim}}^{\text{pres}}$ $\left(= \frac{V_{\text{prim}}^{\text{final}}}{V_{\text{prim}}^{\text{ini}}}\right)$. Secondly, for simplicity, we assume that the preservation factor $\chi_{\text{prim}}^{\text{pres}}$ remains roughly constant for a given set of physical parameters of primordial material (viscosity contrast $\lambda_{\text{prim}}$ and intrinsic buoyancy number $B$). Moreover, each experiment (set of $\lambda_{\text{prim}}$ and $B$), we set $\chi_{\text{prim}}^{\text{pres}}$ equal to the resulting final primordial heterogeneity volume ( $\chi_{\text{prim}}^{\text{LM}}$ in Table C1) fraction from the same experiment in model suite 1. With these assumptions, we rewrite eq. (3) as:

$$f_{\text{bs,amb}}^{\text{final}} = \frac{f_{\text{bs,amb}}^{\text{ini}} \cdot \left(V_{\text{mantle}} - V_{\text{prim}}^{\text{ini}}\right) + c_{\text{bs,prim}} \cdot \left(1 - \chi_{\text{prim}}^{\text{pres}}\right) \cdot V_{\text{prim}}^{\text{ini}}}{V_{\text{mantle}} - \chi_{\text{prim}}^{\text{pres}} \cdot V_{\text{prim}}^{\text{ini}}} \tag{A2}$$

Alternatively, we can make $V_{\text{prim}}^{\text{ini}}$ dependent on $f_{\text{bs,amb}}^{\text{final}}$ by rewriting eq. (A2):

$$f_{\text{bs,amb}}^{\text{final}} \cdot V_{\text{mantle}} - f_{\text{bs,amb}}^{\text{final}} \cdot \chi_{\text{prim}}^{\text{pres}} \cdot V_{\text{prim}}^{\text{ini}} = f_{\text{bs,amb}}^{\text{ini}} \cdot V_{\text{mantle}} - f_{\text{bs,amb}}^{\text{ini}} \cdot V_{\text{prim}}^{\text{ini}} + c_{\text{bs,prim}} \cdot \left(1 - \chi_{\text{prim}}^{\text{pres}}\right) \cdot V_{\text{prim}}^{\text{ini}} \tag{A3}$$

$$f_{\text{bs,amb}}^{\text{ini}} \cdot V_{\text{prim}}^{\text{ini}} - f_{\text{bs,amb}}^{\text{final}} \cdot \chi_{\text{prim}}^{\text{pres}} \cdot V_{\text{prim}}^{\text{ini}} - c_{\text{bs,prim}} \cdot \left(1 - \chi_{\text{prim}}^{\text{pres}}\right) \cdot V_{\text{prim}}^{\text{ini}} = f_{\text{bs,amb}}^{\text{ini}} \cdot V_{\text{mantle}} - f_{\text{bs,amb}}^{\text{final}} \cdot V_{\text{mantle}} \tag{A4}$$

$$V_{\text{prim}}^{\text{ini}} \left(f_{\text{bs,ambt}}^{\text{ini}} - f_{\text{bs,amb}}^{\text{final}} \cdot \chi_{\text{prim}}^{\text{pres}} - c_{\text{bs,prim}} \cdot \left(1 - \chi_{\text{prim}}^{\text{pres}}\right)\right) = V_{\text{mantle}} \left(f_{\text{bs,amb}}^{\text{ini}} - f_{\text{bs,amb}}^{\text{final}}\right) \tag{A5}$$

leading to the expression for $V_{\text{prim}}^{\text{ini}}$:

$$V_{\text{prim}}^{\text{ini}} = V_{\text{mantle}} \frac{f_{\text{bs,amb}}^{\text{final}} - f_{\text{bs,amb}}^{\text{ini}}}{c_{\text{bs,prim}} \cdot \left(1 - \chi_{\text{prim}}^{\text{pres}}\right) + \chi_{\text{prim}}^{\text{pres}} \cdot f_{\text{bs,amb}}^{\text{final}} - f_{\text{bs,amb}}^{\text{ini}}} \tag{A6}$$

We use eq. (A6) and set the target ambient mantle composition on $f_{\text{bs,amb}}^{\text{final}} = 0.25$ to find $V_{\text{prim}}^{\text{ini}}$ (and hence $D_{\text{prim}}$) for each case. Note that since we use spherical annulus geometry, all volumes scale as in 3D spherical geometry (Hernlund and Tackley, 2008).





Following this approach, all models in model suite 2 display a final ambient mantle composition of $f_{\text{bs,amb}}^{\text{final}}$ = 0.24-0.26 (Table C2) after 4.5 Gyr model time. This outcome demonstrates that the above assumptions are valid within reasonable margin.

**Appendix B: Plate-like behaviour in numerical models**

As in Tackley (2000), we measure plateness $P$ (the degree to which surface deformation is localized) and mobility $M$ (the extend to which the lithosphere is able to move). Plateness is defined as:

$$P = 1 - f_{80}/f_{80,\text{iso}} \tag{B1}$$

were $f_{80}$ corresponds to the proportion of the surface that localises 80% of the total deformation and the value of $f_{80,\text{iso}}$ for an isoviscous model (about 0.6 for models with Rayleigh number $> 10^6$, Tackley, 2000). The mobility $M$ is the ratio of the root mean-square (RMS) surface velocity to RMS velocity in the whole domain. Plate-like behaviour occurs when $P$ is close to 1 and $M$ is close to or larger than 1 (Tackley, 2000).

In Figure B1, the temporal evolution of plateness $P$ and mobility $M$ is plotted for a representative model for each identified mantle heterogeneity style. The models correspond to the ones presented in Figures 3 and 4. Many of the "partial heterogeneity preservation" models (regime *III*) display plate tectonic behaviour after the onset of whole-mantle convection (i.e., after the compositional overturn).




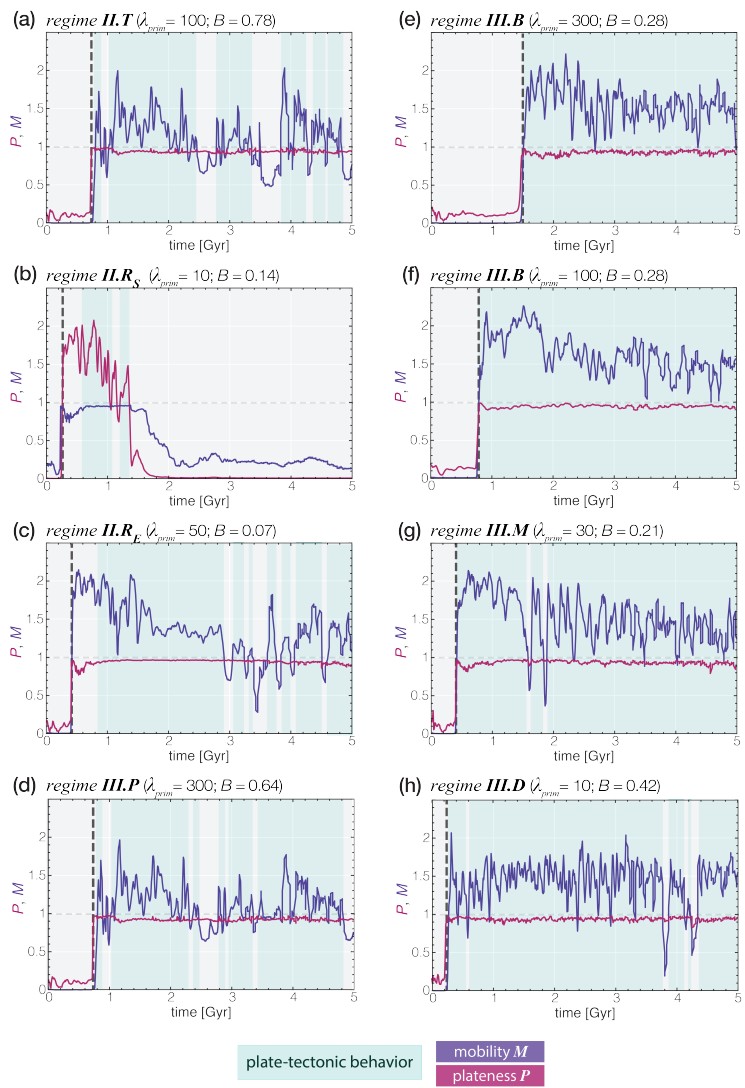

**Figure B1.** Plateness $P$ and mobility $M$ through time for the selected cases of each regime identified in the first model suite. The black dotted line indicates the onset of the compositional overturn. Plate-tectonic behavior (in line with Tackley, 2000) intervals are marked in shaded blue.





## Appendix C: Summary of numerical experiments

Table C1: Model parameter summary and output quantities, averaged between 4.25 and 4.5 Gyr of model evolution. Models in this table belong to the first explored model suite (see text). Plateness $P$ and mobility $M$ from Tackley (2000); * denotes the reference model.

| Model | $\lambda_{prim}$ | $B\,(Mg\#)$ | $D_{prim}$ [km] | $t_{ot}$ [Gyr] | $P$ | $M$ | $T_{mean}$ [K] | $\eta_{av}$ [Pa·s] | $f_{bs,amb}^{final}$ | $\chi_{prim}^{LM}$ [v%LM] | $\chi_{ROC}^{LM}$ [v%LM] | regime |
|---|---|---|---|---|---|---|---|---|---|---|---|---|
| $M_{1a}$ | 1 | 0.07 (0.9) | 2230 | 0.12 | 0.98 | 0.90 | 1937 | $5.1\cdot10^{24}$ | 0.32 | 0.00 | 10.54 | II.R$_S$ |
| $M_{1b}$ | 1 | 0.14 (0.89) | 2230 | 0.32 | 0.96 | 1.16 | 1969 | $4.9\cdot10^{24}$ | 0.32 | 0.00 | 11.31 | II.R$_S$ |
| $M_{1c}$ | 1 | 0.21 (0.89) | 2230 | 0.70 | 0.99 | 0.72 | 1993 | $5.5\cdot10^{24}$ | 0.32 | 0.00 | 12.99 | II.R$_S$ |
| $M_{1d}$ | 1 | 0.28 (0.88) | 2230 | 0.85 | 0.16 | 0.00 | 2066 | $6\cdot10^{24}$ | 0.32 | 0.00 | 13.76 | II.R$_S$ |
| $M_{1e}$ | 1 | 0.35 (0.88) | 2230 | 0.80 | 0.20 | 0.00 | 2052 | $6.1\cdot10^{24}$ | 0.32 | 0.00 | 14.12 | II.R$_S$ |
| $M_{1f}$ | 1 | 0.42 (0.87) | 2230 | 0.98 | 0.15 | 0.00 | 2067 | $5.9\cdot10^{24}$ | 0.32 | 0.00 | 14.65 | II.R$_S$ |
| $M_{1g}$ | 1 | 0.50 (0.87) | 2230 | 0.96 | 0.96 | 1.07 | 2006 | $5\cdot10^{24}$ | 0.32 | 0.03 | 11.81 | II.R$_S$ |
| $M_{1h}$ | 1 | 0.57 (0.86) | 2230 | 0.92 | 0.98 | 0.77 | 2044 | $4.5\cdot10^{24}$ | 0.32 | 0.20 | 11.43 | II.R$_S$ |
| $M_{1i}$ | 1 | 0.64 (0.86) | 2230 | 1.08 | 1.16 | 0.00 | 2334 | $6.1\cdot10^{24}$ | 0.23 | 77.26 | 7.82 | II.R$_S$ |
| $M_{1j}$ | 1 | 0.71 (0.85) | 2230 | 1.00 | 0.96 | 0.04 | 2290 | $4.9\cdot10^{24}$ | 0.24 | 69.54 | 8.70 | II.R$_S$ |
| $M_{1k}$ | 1 | 0.78 (0.85) | 2230 | 1.7; 3.1 | 0.97 | 0.43 | 2335 | $2.8\cdot10^{24}$ | 0.23 | 77.23 | 7.65 | II.R$_S$ |
| $M_{10a}$ | 10 | 0.07 (0.9) | 2230 | 0.24 | 0.25 | 0.01 | 1975 | $7.0\cdot10^{24}$ | 0.32 | 0.0 | 11.2 | II.R$_S$ |
| $M_{10b}$ | 10 | 0.14 (0.89) | 2230 | 0.24 | 0.27 | 0.01 | 1967 | $6.9\cdot10^{24}$ | 0.32 | 0.1 | 10.5 | II.R$_S$ |
| $M_{10c}$ | 10 | 0.21 (0.89) | 2230 | 0.25 | 0.96 | 1.35 | 1963 | $9.8\cdot10^{23}$ | 0.31 | 1.8 | 6.4 | III.M |
| $M_{10d}$ | 10 | 0.28 (0.88) | 2230 | 0.27 | 0.96 | 1.31 | 2061 | $9.0\cdot10^{23}$ | 0.31 | 2.05 | 5.2 | III.M |
| $M_{10e}$ | 10 | 0.35 (0.88) | 2230 | 0.27 | 0.94 | 1,52 | 2084 | $5.6\cdot10^{23}$ | 0.30 | 2.2 | 5.3 | III.M |
| $M_{10f}$ | 10 | 0.42 (0.87) | 2230 | 0.28 | 0.92 | 1.18 | 2133 | $7.9\cdot10^{23}$ | 0.29 | 2.6 | 5.8 | III.D |
| $M_{10g}$ | 10 | 0.50 (0.87) | 2230 | 0.38 | 0.93 | 1.51 | 2179 | $7.8\cdot10^{23}$ | 0.29 | 5.3 | 5.0 | III.D |
| $M_{10h}$ | 10 | 0.57 (0.86) | 2230 | 0.4 | 0.92 | 1.46 | 2205 | $6.5\cdot10^{23}$ | 0.28 | 7.0 | 4.6 | III.D |
| $M_{10i}$ | 10 | 0.64 (0.86) | 2230 | 0.88 | 0.91 | 1.25 | 2184 | $6.1\cdot10^{23}$ | 0.23 | 31.3 | 4.2 | III.P |
| $M_{10j}$ | 10 | 0.71 (0.85) | 2230 | 1.17 | 0.91 | 1.2 | 2190 | $4.9\cdot10^{23}$ | 0.24 | 33.6 | 3.6 | III.P |
| $M_{10k}$ | 10 | 0.78 (0.85) | 2230 | - | 0.92 | 0.98 | 2235 | $3.8\cdot10^{23}$ | 0.23 | 47.7 | 2.02 | II.T |
| $M_{30a}$ | 30 | 0.07 (0.9) | 2230 | 0.40 | 0.95 | 1.53 | 1930 | $3.2\cdot10^{24}$ | 0.32 | 1.06 | 9.25 | II.R$_E$ |
| $M_{30b}$ | 30 | 0.14 (0.89) | 2230 | 0.40 | 0.95 | 1.22 | 1934 | $2\cdot10^{24}$ | 0.32 | 2.00 | 7.70 | II.R$_E$ |
| $M_{30c}$ | 30 | 0.21 (0.89) | 2230 | 0.44 | 0.94 | 1.36 | 1992 | $6.4\cdot10^{23}$ | 0.32 | 2.82 | 4.83 | III.M |
| $M_{30d}$ | 30 | 0.28 (0.88) | 2230 | 0.50 | 0.94 | 1.11 | 2078 | $6.7\cdot10^{23}$ | 0.31 | 5.02 | 4.22 | III.B |
| $M_{30e}$ | 30 | 0.35 (0.88) | 2230 | 0.65 | 0.97 | 1.33 | 2130 | $6.1\cdot10^{23}$ | 0.31 | 9.24 | 4.05 | III.B |
| $M_{30f}$ | 30 | 0.42 (0.87) | 2230 | 0.80 | 0.95 | 1.44 | 2150 | $5.5\cdot10^{23}$ | 0.31 | 13.26 | 3.89 | III.B |
| $M_{30g}$ | 30 | 0.50 (0.87) | 2230 | 1.00 | 0.95 | 1.63 | 2177 | $4.7\cdot10^{23}$ | 0.30 | 23.81 | 3.41 | III.B |
| $M_{30h}$ | 30 | 0.57 (0.86) | 2230 | 1.22 | 0.92 | 1.55 | 2180 | $4.2\cdot10^{23}$ | 0.30 | 28.47 | 2.24 | III.B |
| $M_{30i}$ | 30 | 0.64 (0.86) | 2230 | 1.48 | 0.94 | 1.40 | 2103 | $6.8\cdot10^{23}$ | 0.29 | 32.84 | 3.03 | III.P |
| $M_{30j}$ | 30 | 0.71 (0.85) | 2230 | 1.70 | 0.91 | 1.41 | 2110 | $1.4\cdot10^{24}$ | 0.26 | 56.01 | 2.96 | III.P |
| $M_{30k}$ | 30 | 0.78 (0.85) | 2230 | - | 0.92 | 1.04 | 2168 | $2.3\cdot10^{24}$ | 0.24 | 70.32 | 1.79 | II.T |
| $M_{50a}$ | 50 | 0.07 (0.90) | 2230 | 0.58 | 0.96 | 1.38 | 1955 | $3.4\cdot10^{24}$ | 0.32 | 1.92 | 7.67 | II.R$_E$ |
| $M_{50b}$ | 50 | 0.14 (0.89) | 2230 | 0.58 | 0.95 | 1.22 | 1936 | $2\cdot10^{24}$ | 0.31 | 4.70 | 7.03 | II.R$_E$ |





| $M_{50c}$ | 50 | 0.21 (0.89) | 2230 | 0.57 | 0.93 | 1.62 | 1941 | $1.2\cdot10^{24}$ | 0.31 | 4.80 | 6.05 | III.M |
|---|---|---|---|---|---|---|---|---|---|---|---|---|
| $M_{50d}$ | 50 | 0.28 (0.88) | 2230 | 0.58 | 0.94 | 1.36 | 2116 | $5.5\cdot10^{23}$ | 0.31 | 11.31 | 3.46 | III.B |
| $M_{50e}$ | 50 | 0.35 (0.88) | 2230 | 0.62 | 0.96 | 1.31 | 2142 | $6.2\cdot10^{23}$ | 0.30 | 17.50 | 4.25 | III.B |
| $M_{50f}$ | 50 | 0.42 (0.87) | 2230 | 0.62 | 0.95 | 1.40 | 2177 | $5.5\cdot10^{23}$ | 0.31 | 14.33 | 3.33 | III.B |
| $M_{50g}$ | 50 | 0.50 (0.87) | 2230 | 0.65 | 0.95 | 1.33 | 2127 | $5.3\cdot10^{23}$ | 0.30 | 18.18 | 2.85 | III.B |
| $M_{50h}$ | 50 | 0.57 (0.86) | 2230 | 0.75 | 0.93 | 1.45 | 2132 | $4.6\cdot10^{23}$ | 0.30 | 20.64 | 1.99 | III.B |
| $M_{50i}$ | 50 | 0.64 (0.86) | 2230 | 1.20 | 0.92 | 1.47 | 2105 | $7.7\cdot10^{23}$ | 0.28 | 40.47 | 2.58 | III.P |
| $M_{50j}$ | 50 | 0.71 (0.85) | 2230 | 1.45 | 0.91 | 1.11 | 2043 | $2.2\cdot10^{24}$ | 0.26 | 58.18 | 3.10 | III.P |
| $M_{50k}$ | 50 | 0.78 (0.85) | 2230 | - | 0.94 | 1.17 | 2155 | $2.4\cdot10^{24}$ | 0.23 | 74.46 | 1.69 | II.T |
| $M_{100a}$ | 100 | 0.07 (0.90) | 2230 | 0.89 | 0.94 | 1.43 | 1935 | $2.5\cdot10^{24}$ | 0.31 | 4.24 | 7.09 | II.R$_E$ |
| $M_{100b}$ | 100 | 0.14 (0.89) | 2230 | 0.90 | 0.96 | 1.20 | 1960 | $2.2\cdot10^{24}$ | 0.31 | 7.28 | 5.50 | III.B |
| $M_{100c}$ | 100 | 0.21 (0.89) | 2230 | 0.90 | 0.94 | 1.36 | 2085 | $5.3\cdot10^{23}$ | 0.31 | 12.11 | 3.09 | III.B |
| $M_{100d}$* | **100** | **0.28 (0.88)** | **2230** | **0.90** | **0.94** | **1.45** | **2105** | **$4.9\cdot10^{23}$** | **0.31** | **15.10** | **3.41** | **III.B** |
| $M_{100e}$ | 100 | 0.35 (0.88) | 2230 | 0.92 | 0.94 | 1.38 | 2142 | $5.1\cdot10^{23}$ | 0.30 | 18.55 | 3.34 | III.B |
| $M_{100f}$ | 100 | 0.42 (0.87) | 2230 | 0.95 | 0.95 | 1.75 | 2156 | $5.3\cdot10^{23}$ | 0.30 | 20.92 | 3.11 | III.B |
| $M_{100g}$ | 100 | 0.50 (0.87) | 2230 | 1.07 | 0.93 | 1.50 | 2114 | $3.8\cdot10^{23}$ | 0.31 | 15.94 | 2.52 | III.B |
| $M_{100h}$ | 100 | 0.57 (0.86) | 2230 | 1.20 | 0.94 | 1.43 | 2109 | $5.2\cdot10^{23}$ | 0.30 | 18.04 | 1.91 | III.B |
| $M_{100i}$ | 100 | 0.64 (0.86) | 2230 | 1.65 | 0.90 | 1.56 | 1957 | $1.5\cdot10^{24}$ | 0.29 | 34.88 | 3.50 | III.P |
| $M_{100j}$ | 100 | 0.71 (0.85) | 2230 | 1.70 | 0.93 | 1.41 | 2009 | $2.9\cdot10^{24}$ | 0.27 | 55.51 | 2.74 | III.P |
| $M_{100k}$ | 100 | 0.78 (0.85) | 2230 | - | 0.94 | 1.20 | 2152 | $2.6\cdot10^{24}$ | 0.22 | 78.45 | 1.54 | II.T |
| $M_{300a}$ | 300 | 0.07 (0.90) | 2230 | 1.48 | 0.92 | 1.56 | 1949 | $1.8\cdot10^{24}$ | 0.31 | 7.79 | 5.30 | III.B |
| $M_{300b}$ | 300 | 0.14 (0.89) | 2230 | 1.48 | 0.96 | 1.37 | 1965 | $1.3\cdot10^{24}$ | 0.31 | 9.10 | 5.22 | III.B |
| $M_{300c}$ | 300 | 0.21 (0.89) | 2230 | 1.48 | 0.94 | 1.45 | 2048 | $6.4\cdot10^{23}$ | 0.31 | 15.32 | 3.19 | III.B |
| $M_{300d}$ | 300 | 0.28 (0.88) | 2230 | 1.48 | 0.94 | 1.42 | 2178 | $5.7\cdot10^{23}$ | 0.30 | 22.88 | 2.51 | III.B |
| $M_{300e}$ | 300 | 0.35 (0.88) | 2230 | 1.50 | 0.95 | 1.69 | 2161 | $5.4\cdot10^{23}$ | 0.30 | 23.79 | 2.13 | III.B |
| $M_{300f}$ | 300 | 0.42 (0.87) | 2230 | 1.50 | 0.92 | 1.57 | 2084 | $5\cdot10^{23}$ | 0.30 | 25.86 | 3.02 | III.B |
| $M_{300g}$ | 300 | 0.50 (0.87) | 2230 | 1.50 | 0.90 | 1.48 | 2021 | $4.9\cdot10^{23}$ | 0.30 | 25.48 | 2.74 | III.B |
| $M_{300h}$ | 300 | 0.57 (0.86) | 2230 | 1.52 | 0.90 | 1.59 | 1936 | $8.3\cdot10^{23}$ | 0.30 | 27.34 | 4.16 | III.P |
| $M_{300i}$ | 300 | 0.64 (0.86) | 2230 | 1.56 | 0.92 | 1.81 | 1935 | $3\cdot10^{24}$ | 0.28 | 43.00 | 3.81 | III.P |
| $M_{300j}$ | 300 | 0.71 (0.85) | 2230 | - | 0.92 | 1.19 | 2022 | $3.6\cdot10^{24}$ | 0.25 | 63.84 | 2.76 | II.T |
| $M_{300k}$ | 300 | 0.78 (0.85) | 2230 | - | 0.94 | 1.41 | 2177 | $3.9\cdot10^{24}$ | 0.21 | 85.00 | 0.66 | II.T |
| $M_{500a}$ | 500 | 0.07 (0.90) | 2230 | 2.00 | 0.96 | 1.38 | 1935 | $2\cdot10^4$ | 0.31 | 8.57 | 5.56 | III.B |
| $M_{500b}$ | 500 | 0.14 (0.89) | 2230 | 2.00 | 0.94 | 1.49 | 1979 | $9.6\cdot10^{23}$ | 0.31 | 10.07 | 3.77 | III.B |
| $M_{500c}$ | 500 | 0.21 (0.89) | 2230 | 2.00 | 0.93 | 1.41 | 2082 | $5.3\cdot10^{23}$ | 0.31 | 17.22 | 3.02 | III.B |
| $M_{500d}$ | 500 | 0.28 (0.88) | 2230 | 2.00 | 0.93 | 1.28 | 2091 | $5.9\cdot10^{23}$ | 0.30 | 22.09 | 3.19 | III.B |
| $M_{500e}$ | 500 | 0.35 (0.88) | 2230 | 2.00 | 0.94 | 1.53 | 2139 | $5.4\cdot10^{23}$ | 0.30 | 24.69 | 2.45 | III.B |
| $M_{500f}$ | 500 | 0.42 (0.87) | 2230 | 2.03 | 0.92 | 1.57 | 2062 | $7.9\cdot10^{23}$ | 0.29 | 31.85 | 3.03 | III.B |
| $M_{500g}$ | 500 | 0.50 (0.87) | 2230 | 2.05 | 0.93 | 1.51 | 2001 | $1\cdot10^{24}$ | 0.29 | 34.48 | 3.62 | III.P |
| $M_{500h}$ | 500 | 0.57 (0.86) | 2230 | 2.10 | 0.91 | 1.64 | 1989 | $1.9\cdot10^{24}$ | 0.28 | 46.68 | 3.60 | III.P |
| $M_{500i}$ | 500 | 0.64 (0.86) | 2230 | 2.10 | 0.92 | 1.88 | 2061 | $3.6\cdot10^{24}$ | 0.24 | 69.70 | 1.59 | III.P |
| $M_{500j}$ | 500 | 0.71 (0.85) | 2230 | - | 0.94 | 1.59 | 1084 | $5\cdot10^{23}$ | 0.23 | 75.01 | 1.66 | II.T |
| $M_{500k}$ | 500 | 0.78 (0.85) | 2230 | - | 0.93 | 1.23 | 2167 | $5\cdot10^{23}$ | 0.20 | 86.19 | 0.54 | II.T |





Table C2: Model parameter summary and output quantities of models in the second model suite (different $D_{\mathrm{prim}}$, see text), averaged between 4.25 and 4.5 Gyr of model evolution. Plateness $P$ and mobility $M$ from Tackley (2000).

| Model | $\lambda_{\mathrm{prim}}$ | $B\,(Mg\#)$ | $D_{\mathrm{prim}}$ [km] | $t_{\mathrm{ot}}$ [Gyr] | $P$ | $M$ | $T_{\mathrm{mean}}$ [K] | $\eta_{\mathrm{av}}$ [Pa·s] | $f_{\mathrm{bs,amb}}^{\mathrm{final}}$ | $\chi_{\mathrm{prim}}^{\mathrm{pres}}$ [v%$_{\mathrm{prim}}^{\mathrm{ini}}$] | $\chi_{\mathrm{prim}}^{\mathrm{LM}}$ [v%$_{\mathrm{LM}}$] | $\chi_{\mathrm{ROC}}^{\mathrm{LM}}$ [v%$_{\mathrm{LM}}$] | regime |
|---|---|---|---|---|---|---|---|---|---|---|---|---|---|
| $M_{10\mathrm{aD}}$ | 10 | 0.07 (0.90) | 1568 | 0.28 | 0.93 | 1.49 | 1935 | $2.1\cdot10^{24}$ | 0.26 | 1.7 | 1.0 | 6.9 | III.M |
| $M_{10\mathrm{bD}}$ | 10 | 0.14 (0.89) | 1568 | 0.30 | 0.95 | 1.49 | 1984 | $8.1\cdot10^{23}$ | 0.26 | 2.5 | 1.5 | 5.7 | III.M |
| $M_{10\mathrm{cD}}$ | 10 | 0.21 (0.89) | 1580 | 0.32 | 0.92 | 1.52 | 2011 | $1.3\cdot10^{24}$ | 0.25 | 2.6 | 1.6 | 6.1 | III.M |
| $M_{10\mathrm{dD}}$ | 10 | 0.28 (0.88) | 1582 | 0.32 | 0.95 | 1.43 | 2021 | $1.1\cdot10^{24}$ | 0.25 | 2.1 | 1.3 | 7.1 | III.M |
| $M_{10\mathrm{eD}}$ | 10 | 0.35 (0.88) | 1583 | 0.34 | 0.97 | 1.40 | 2087 | $9.3\cdot10^{23}$ | 0.25 | 3.0 | 1.8 | 6.0 | III.D |
| $M_{10\mathrm{fD}}$ | 10 | 0.42 (0.87) | 1586 | 0.35 | 0.94 | 1.13 | 2115 | $8.5\cdot10^{23}$ | 0.25 | 3.3 | 2.0 | 5.7 | III.D |
| $M_{10\mathrm{gD}}$ | 10 | 0.50 (0.87) | 1603 | 0.33 | 0.97 | 1.24 | 2186 | $9.3\cdot10^{23}$ | 0.26 | 3.9 | 2.4 | 4.2 | III.D |
| $M_{30\mathrm{aD}}$ | 30 | 0.07 (0.90) | 1574 | 0.51 | 0.94 | 1.38 | 1999 | $6.9\cdot10^{23}$ | 0.25 | 1.84 | 1.11 | 3.96 | III.M |
| $M_{30\mathrm{bD}}$ | 30 | 0.14 (0.89) | 1569 | 0.52 | 0.94 | 1.42 | 2063 | $5.7\cdot10^{23}$ | 0.25 | 1.7 | 1.02 | 4.45 | III.M |
| $M_{30\mathrm{cD}}$ | 30 | 0.21 (0.89) | 1583 | 0.6 | 0.94 | 1.56 | 2063 | $9.0\cdot10^{23}$ | 0.25 | 6.64 | 4.04 | 5.29 | III.B |
| $M_{30\mathrm{dD}}$ | 30 | 0.28 (0.88) | 1603 | 0.54 | 0.93 | 1.61 | 2090 | $8.7\cdot10^{23}$ | 0.25 | 8.14 | 5.04 | 5.82 | III.B |
| $M_{30\mathrm{eD}}$ | 30 | 0.35 (0.88) | 1627 | 0.57 | 0.95 | 1.51 | 2150 | $8.2\cdot10^{23}$ | 0.25 | 3.81 | 2.41 | 5.50 | III.B |
| $M_{30\mathrm{fD}}$ | 30 | 0.42 (0.87) | 1656 | 0.68 | 0.96 | 1.57 | 2202 | $5.8\cdot10^{23}$ | 0.25 | 4.91 | 3.18 | 4.23 | III.D |
| $M_{30\mathrm{gD}}$ | 30 | 0.50 (0.87) | 1728 | 0.85 | 0.94 | 1.43 | 2214 | $4.6\cdot10^{23}$ | 0.26 | 5.2 | 3.66 | 2.83 | III.D |
| $M_{50\mathrm{aD}}$ | 50 | 0.07 (0.90) | 1583 | 0.54 | 0.93 | 1.54 | 2034 | $5.6\cdot10^{23}$ | 0.25 | 3.76 | 2.29 | 3.61 | III.M |
| $M_{50\mathrm{bD}}$ | 50 | 0.14 (0.89) | 1594 | 0.53 | 0.93 | 1.59 | 2085 | $5.1\cdot10^{23}$ | 0.25 | 7.73 | 4.75 | 5.30 | III.B |
| $M_{50\mathrm{cD}}$ | 50 | 0.21 (0.89) | 1607 | 0.55 | 0.94 | 1.26 | 2110 | $7.3\cdot10^{23}$ | 0.25 | 13 | 8.07 | 4.82 | III.B |
| $M_{50\mathrm{dD}}$ | 50 | 0.28 (0.88) | 1617 | 0.58 | 0.96 | 1.51 | 2130 | $7.6\cdot10^{23}$ | 0.25 | 7.56 | 4.74 | 5.82 | III.B |
| $M_{50\mathrm{eD}}$ | 50 | 0.35 (0.88) | 1695 | 0.64 | 0.93 | 1.48 | 2171 | $5.5\cdot10^{23}$ | 0.25 | 13.2 | 8.83 | 5.50 | III.B |
| $M_{50\mathrm{fD}}$ | 50 | 0.42 (0.87) | 1656 | 0.72 | 0.95 | 1.37 | 2210 | $5.1\cdot10^{23}$ | 0.25 | 12.82 | 8.31 | 3.39 | III.D |
| $M_{50\mathrm{gD}}$ | 50 | 0.50 (0.87) | 1728 | 1 | 0.95 | 1.54 | 2192 | $4.7\cdot10^{23}$ | 0.25 | 15.1 | 10.4 | 2.01 | III.D |
| $M_{100\mathrm{aD}}$ | 100 | 0.07 (0.90) | 1590 | 0.6 | 0.93 | 1.43 | 2036 | $6.1\cdot10^{23}$ | 0.25 | 9.21 | 5.64 | 3.96 | III.B |
| $M_{100\mathrm{bD}}$ | 100 | 0.14 (0.89) | 1615 | 0.58 | 0.95 | 1.51 | 2105 | $6.3\cdot10^{23}$ | 0.24 | 16.15 | 10.1 | 4.05 | III.B |
| $M_{100\mathrm{cD}}$ | 100 | 0.21 (0.89) | 1619 | 0.62 | 0.96 | 1.6 | 2161 | $5.7\cdot10^{23}$ | 0.24 | 23.7 | 14.88 | 4.44 | III.B |
| $M_{100\mathrm{dD}}$ | 100 | 0.28 (0.88) | 1650 | 0.68 | 0.95 | 1.3 | 2212 | $6.2\cdot10^{23}$ | 0.24 | 23.7 | 15.27 | 3.42 | III.B |
| $M_{100\mathrm{eD}}$ | 100 | 0.35 (0.88) | 1706 | 0.74 | 0.94 | 1.44 | 2222 | $4.9\cdot10^{23}$ | 0.25 | 15.83 | 10.7 | 2.50 | III.B |
| $M_{100\mathrm{fD}}$ | 100 | 0.42 (0.87) | 1717 | 0.85 | 0.93 | 1.51 | 2204 | $4.1\cdot10^{23}$ | 0.25 | 18.62 | 12.7 | 2.17 | III.B |
| $M_{100\mathrm{gD}}$ | 100 | 0.50 (0.87) | 1657 | 1.4 | 0.95 | 1.54 | 2189 | $5.4\cdot10^{23}$ | 0.24 | 32.2 | 20.9 | 1.33 | III.B |
| $M_{300\mathrm{aD}}$ | 300 | 0.07 (0.90) | 1617 | 0.68 | 0.94 | 1.19 | 2105 | $5.6\cdot10^{23}$ | 0.25 | 13.29 | 8.33 | 2.96 | III.B |
| $M_{300\mathrm{bD}}$ | 300 | 0.14 (0.89) | 1636 | 0.76 | 0.94 | 1.66 | 2165 | $5.0\cdot10^{23}$ | 0.24 | 32.2 | 20.6 | 3.02 | III.B |
| $M_{300\mathrm{cD}}$ | 300 | 0.21 (0.89) | 1678 | 0.8 | 0.95 | 1.41 | 2185 | $6.4\cdot10^{23}$ | 0.24 | 35.6 | 23.5 | 3.14 | III.B |
| $M_{300\mathrm{dD}}$ | 300 | 0.28 (0.88) | 1717 | 0.92 | 0.94 | 1.61 | 2194 | $6.1\cdot10^{23}$ | 0.24 | 35.5 | 24.2 | 2.51 | III.B |
| $M_{300\mathrm{eD}}$ | 300 | 0.35 (0.88) | 1743 | 1.18 | 0.94 | 1.53 | 2167 | $4.2\cdot10^{23}$ | 0.24 | 35.31 | 24.6 | 1.96 | III.B |
| $M_{300\mathrm{fD}}$ | 300 | 0.42 (0.87) | 1768 | 1.38 | 0.91 | 1.69 | 2105 | $5.6\cdot10^{23}$ | 0.25 | 32.07 | 22.8 | 1.33 | III.B |
| $M_{300\mathrm{gD}}$ | 300 | 0.50 (0.87) | 1762 | 1.64 | 0.93 | 1.68 | 2025 | $6.1\cdot10^{23}$ | 0.24 | 36.18 | 25.6 | 1.37 | III.P |
| $M_{500\mathrm{aD}}$ | 500 | 0.07 (0.90) | 1621 | 0.8 | 0.95 | 1.35 | 2068 | $6.3\cdot10^{23}$ | 0.25 | 12.27 | 7.72 | 3.46 | III.B |
| $M_{500\mathrm{bD}}$ | 500 | 0.14 (0.89) | 1635 | 0.9 | 0.94 | 1.31 | 2127 | $5.7\cdot10^{23}$ | 0.24 | 25.3 | 16.1 | 2.66 | III.B |
| $M_{500\mathrm{cD}}$ | 500 | 0.21 (0.89) | 1699 | 1.02 | 0.95 | 1.69 | 2180 | $5.3\cdot10^{23}$ | 0.23 | 41.67 | 28.0 | 2.79 | III.B |





| | | | | | | | | | | | | | |
|---|---|---|---|---|---|---|---|---|---|---|---|---|---|
| $M_{500dD}$ | 500 | 0.28 (0.88) | 1728 | 1.32 | 0.94 | 1.56 | 2172 | $6.3 \cdot 10^{23}$ | 0.24 | 38.94 | 26.8 | 2.53 | III.B |
| $M_{500eD}$ | 500 | 0.35 (0.88) | 1762 | 1.48 | 0.92 | 1.51 | 2150 | $4.9 \cdot 10^{23}$ | 0.24 | 37.45 | 26.5 | 1.85 | III.B |
| $M_{500fD}$ | 500 | 0.42 (0.87) | 1840 | 1.7 | 0.92 | 1.54 | 2084 | $5.7 \cdot 10^{23}$ | 0.25 | 38.52 | 29.0 | 2.27 | III.B |
| $M_{500gD}$ | 500 | 0.50 (0.87) | 1844 | 2 | 0.93 | 1.71 | 2012 | $9.1 \cdot 10^{23}$ | 0.24 | 43.96 | 33.2 | 1.37 | III.P |
| $M_{50dD}^{B0}$ | 50 | 0.28 (0.88) | 1539 | 0.55 | 0.95 | 1.62 | 2108 | $6.2 \cdot 10^{23}$ | 0.30 | 42 | 26.3 | 4.7 | III.B |
| $M_{50dD}^{B1}$ | 50 | 0.28 (0.88) | 1514 | 0.57 | 0.94 | 1.44 | 2059 | $1.2 \cdot 10^{24}$ | 0.30 | 37.7 | 23.7 | 6.7 | III.B |
| $M_{100dD}^{B0}$ | 100 | 0.28 (0.88) | 1571 | 0.70 | 0.94 | 1.52 | 2101 | $6.3 \cdot 10^{23}$ | 0.30 | 41.5 | 27.0 | 4.1 | III.B |
| $M_{100dD}^{B1}$ | 100 | 0.28 (0.88) | 1546 | 0.72 | 0.95 | 1.64 | 2038 | $1.1 \cdot 10^{24}$ | 0.30 | 43.4 | 28.2 | 6.3 | III.B |
| $M_{100dD}^{B2}$ | 100 | 0.28 (0.88) | 1521 | 0.76 | 0.92 | 1.80 | 2039 | $1.5 \cdot 10^{24}$ | 0.30 | 40 | 26.1 | 8.7 | III.B |

[B0-2] stands for additional ancient FeO-rich ancient layer on top of the core-mantle boundary, with layer thickness 150 ([B0]), 200 ([B1]) and 250 ([B2]) km, respectively.





*Author contributions.* A.G. designed the study, conducted the experiments, interpreted the results, and prepared this manuscript. M.B. con-
tributed to the study design and interpretation of the results. P.T. designed the 3D thermomechanical code and contributed to results interpre-
tation. All authors collaborated and contributed intellectually to this paper.

*Competing interests.* The authors declare no competing interests.

*Acknowledgements.* This study was funded by the ETH Zürich grant ETH-33 16-1. All numerical simulations were performed on ETH
Zürich's Euler cluster. For 2D visualisation of the models, we used the open-source software ParaView (http://paraview.org). Several per-
ceptually uniform scientific colour maps (Crameri 2018, http://doi.org/10.5281/zenodo.1243862) were used to prevent visual distortion of
the figures. Finally, the open-source Python module StagPy (https://stagpy.readthedocs.io/en/stable/) was used for post-processing of the
numerical data.



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
