# Peer review of "Coupled dynamics and evolution of primordial and recycled heterogeneity in Earth's lower mantle"

_Solid Earth, 2020_

## Referee Comment (RC1) · Craig ONeill (Referee) · 11 Mar 2021

This is a well written and presented paper that covers the mixing of heterogenous mantle reservoirs in Earth evolution. In particular, it articulates clearly the difference between LLSVP reservoirs, and BEAM-like or ribbon-like mid-mantle reservoirs, showing how they could form independently and represent relict mantle of different antiquity, potentially reconciling some geochemical and geophysical anomalies.

I think, on the whole, it's quite publishable. The assumptions and simplifications behind

the modelling are quite clear, and the results and discussions follow clearly from that. I think the presentation (and ordering) of the information could be improved, and I offer some suggestions for that below - mostly around the ordering of the results, presentation of snapshots, background material appearing in the discussion section, etc. It's probably easiest to approach this sequentially, so I'll do that below.

Line 50-54: I note you mention 3D mixing in the discussion later, but I feel it is worthwhile bringing that content into the background here. In particular, Coltice and Schmalz showed the differences between 2 and 3D for high-Ra regimes were not that significant, so you could bring that discussion here. For more realistic convecting systems, you might considering looking at these 2:

O'Neill, C., Debaille, V. and Griffin, W., 2013. Deep earth recycling in the Hadean and constraints on surface tectonics. American Journal of Science, 313(9), pp.912-932.

https://www.ajsonline.org/content/313/9/912.short

O'Neill, C.J. and Zhang, S., 2018. Lateral mixing processes in the Hadean. Journal of Geophysical Research: Solid Earth, 123(8), pp.7074-7089. https://agupubs.onlinelibrary.wiley.com/doi/full/10.1029/2018JB015698

Which I somewhat bashfully suggest as I was involved in them (thus - take it or leave it, but relevant if you want to justify using 2D c/w 3D).

Line 66-67: "robustly predicted in many experiments" - better add a cite here.

Line 69: "a hybrid state" - do you think it is terribly different from marble cake with scale variance?

Line 76: "512.96" looks like a decimal, expand to "512 x 96 cells"

Table 1. The activation volume and energies are the same for the upper mantle (mostly olivine) and bridgmanite (PV). This is a bit of a stretch, and it was worth noting this in the review (although I'm sure nothing new to the senior authors). The energies themselves

are very low compared to lab measurements (c/w Hirth and Kohlstedt, or Karato and Wu for olivine, E~375e3), and the activation volume is appropriate for bridgmanite, but not for olivine. This certainly makes the models easier to run as things converge, but is it realistic? There is some token justification (lower mantle/grain size/stress) but this is not very well developed. I would like to see a little more text on justifying using these parameters properly, and what the potential implications are of this choice (eg. for the upper mantle, it results in very different behaviour near the lithosphere).

L147-148: "density (FeO enrichment)": can you expand on the exact density difference between FeO and bridgmanite in the lower mantle, as this is quite relevant here. It sounds like you are saying FeO is denser - I encourage to revisit this statement carefully (and calculate the density profiles of bridgmanite vs FeO in burnman to see what I mean).

Figure 2: I feel like this figure should come after the figures showing the timeseries of behaviour, which probably feeds into the whole structure of the results. Show the models first, and describe the mixing patterns, and then present a regime diagram of mixing. Showing this first is topsy-turvy and throws the structure of this section off.

Line 252-253: If they are not directly related to the Earth, it does beg the question of whether you want to include them or not. When I read this part, it strikes me as student project (I know, I know, but best avoid that impression in a published paper), where you made them again and went back and had to do them again. I mean, that's how we work, but often a paper is improved by leaving out the bits that are not really relevant to the final message. Could these be an appendix?

L264: initial primordial layer thickness: can you justify this better? why 1844km thick?

L285-287. Could you expand on this more? I'm familiar with the work, so I get what you're saying, but not everyone will. Why exactly does a thinner bridgmanite layer lead to lower melting? Is it a fertility issue? Or dynamic? And maybe expand on why hotter mantle leads to more melting and thus a cooler mantle - those thinking in terms of plate

cooling only will be thrown by this point.

L351-356: Disproportionation of FeO during core formation is the most obvious (and increasingly commonly cited) cause of lower mantle heterogeneity. It's not explicitly mentioned here - I recommend Frost and McCammon (2008) for a summary.

L362-364: This goes off on a tangent here - is it needed? I feel this opens a can of worms you don't have space to really cover.

L375: I think Rhodri thought he could explain the anticorrelation with the thermal (PPV) mechanism.

L382-391: The discussion in generally is a bit more disjointed than a lot of the rest of the paper, and skips from point to point without much continuity. The section in particular is hard to follow, and just really needs a rewrite.

L408-411: Can you explain better what the iron spin transition has to do with bridgmanite structure size/scale? This is not clear.

L432-434: MgO-rich komatiites are very much more dense (at the surface) than basalt - are you talking about after the eclogite transition? If so, this statement needs to be much clearer, as written it is quite wrong. See van Thienen et al (2004) for effect of MgO content on mafic rocks.

L456-457: I think you need to cite the old original Stein and Hoffman (1994) mantle overturn model here: Stein, M. and Hofmann, A.W., 1994. Mantle plumes and episodic crustal growth. Nature, 372(6501), pp.63-68.

L477-479: A disjunct with seismic models seems problematic. It's one thing to point it out, but as it stands you seem to be letting it undermine the whole work. You should really advance a reason for the discrepancy (are the differences - quantitatively - so large in terms of T? Is your core evolving? (No? Then it may overestimate heat flow. Etc etc).

L483-487: Again I think this whole discussion of 2D vs 3D is best nipped in the bud at the start - see previous comment.

L490: Without radioactivity or core evolution, these aren't strictly evolutionary models, they are static steady-state models, started at adiabatic conditions, that are run for long periods of time. Apart from the destabilisation of heterogeneities, there is not much 'evolutionary' in them, and this point needs to be made clearer, and earlier.

L508-510: Again, this statement needs to be couched. The timescales for systems with significant radioactive decay are likely very different from the steady-state systems here. A CMB temperature of 4000K is probably ok for some of the early Earth - it's a little higher than estimated today - but probably underestimates early evolution significantly. I think needs to be tackled more explicitly in the discussion, along the lines: "What can we tell about mixing timescales on the Earth from steady state models? Well, we select TCMB and adiabat values around ballpark (average for this period - acknowledging these values evolved rapidly during this time). Based on this we can say..."

I don't think this is too much overhead, but at least shuts down the critique of the evolutionary aspect missing here.

Ok, that's all from me. Nice job - and good luck!

-CO

---

## Referee Comment (RC2) · Shijie Zhong (Referee) · 18 Mar 2021

General comments:

This is a comprehensive modeling study on mantle composition, structure, and mixing. The study includes >100 cases that cover a large parameter space on viscosity, buoyancy ratio and other parameters. This is a significant modeling effort, even though the models are done 2-D. The study based on the modeling results identifies two main regimes: chemical stratification (regime II) and partial heterogeneity preservation (regime III), and each regime has a number of sub-regimes. All the regimes and

sub-regimes have been proposed in one way or other previously mostly as conceptual models of the mantle, but it is instructive to see that the numerical models reproduce these regimes/sub-regimes for different model parameters. In particular, the study highlights a regime where recycled oceanic crust persists as large piles at the base of the mantle, while primordial material exists as blobs in the mid-mantle. The paper is generally well written, especially considering the complicated nature of the modeling. The study is a useful contribution to the ongoing discussion on mantle composition and structure. However, I do have some concerns that the authors should address.

Specific comments:

1) My main concern is about whether the 2-D models have sufficient numerical resolution to resolve the entrainment process that is so key to some of the model results reported here, e.g., the survival of the primordial materials and formation of the piles above the CMB. The 2-D models with fairly high Rayleigh number ($\sim$10ˆ7) have $\sim$100 grid points in radial direction covering $\sim$3000 km thick mantle. As a comparison, Leng and Zhong (2011) showed that using adaptive mesh refinement method, mesh resolution corresponding to 256 elements in vertical direction is needed to reproduce the entrainment rate by Davaille's analytical prediction for Ra=10ˆ5. We tried unsuccessfully for computing entrainment rate in models with variable viscosity. Zhong and Hager (2003) also used the marker chain method to determine the entrainment rate for a plume with fixed buoyancy on the top of a dense layer, demonstrating the requirement of numerical resolution. Van Keken et al., (1997) in their benchmark paper also showed the difficulty in computing entrainment rate. In Li et al. (2014), their 2-D models only cover the bottom of the mantle with super-high resolution ($\sim$3 km?) to study the recycled crustal material interacting with the basal layer. Based on these studies, I have a good reason to believe that the models here (high Ra and variable viscosity) may have over-predicted the entrainment, but I do not know how much it would affect the overall conclusions. It would be very helpful for the authors to pick a case for a resolution test. To this end, I can see the need for at least two calculations using 200 and 400 grid

points in radial direction, given the high Ra and variable viscosity in their models (both making entrainment more difficult to compute).

2) The manuscript acknowledged in the final sub-section the drawback of ignoring the radioactive heating. For a study like the current one aiming at the long-term thermal and chemical evolution of the mantle, this seems to be a significant deficiency. I think that the authors should acknowledge it in the abstract and conclusions sections.

3) It would be helpful to clarify how the viscosity is computed. Equation (1) gives the viscosity for each composition through the pre-factor, lambda. I suppose that in solving the Stokes' equation, the viscosity is assigned to grid points. Then how are viscosities of different composition in a vicinity of a grid point (or within in a grid) averaged and assigned to the grid?

4) To follow point 3 above on viscosity, it seems that the manuscript focuses on the effect of lambda_prim, while not mentioning much on lambda_LM and lambda_ppv. Also, it appears that lambda_prim controls whether a convection is in a stagnant lid (for a small lambda_prim) regime or mobile-lid regime (for a large lambda_prim). Can the authors provide more insight or explanation as to why lambda_prim could have such an effect? Most previous studies show that for this type of models, large activation energy would lead to stagnant-lid convection, but the current study seems to suggest otherwise. It is unclear to me why lambda_prim has such a power. Along this line, does a large lambda_prim mean a more viscous blob (i.e., high viscosity blob)? Here a high radioactive heating for the primordial material may potentially make a big difference, as it would heat up the blob over time.

5) Regime I reported in Gulcher et al., (2020b) does not exist anymore in the current study, as stated in line 168, and can the authors explain why Regime I does not exist in the current models?

6) In Fig. 7, what prevents the primordial materials from entering the upper mantle? The phase change or compositional buoyancy or both? Some explanation would be

helpful.

7) Lines 441-443, " . . . ongoing debate about whether thermochemical piles are intrinsically stable . . .". More appropriate references may be McNamara and Zhong (2005, Nature) and Zhang et al., (2010, JGR).

Technical Comments:

1) Line 35, delete "to".

2) Line 178, Fig. 4a should 3a.

3) Fig. 3's caption, "regime I" or regime II? Fig. 4's caption, "regime II" or regime III? In general, the regime names are somewhat confusing. Another example is in lines 200 (the title of subsection 3.1.2) and 201: the sub-title in line 200 says "regime III", but line 201 says "2nd regime".

4) Fig. 4a, B=64 or 0.64?

5) Fig. 10, BEAMS? What does it stand for?

---

## Author Comment (AC1) · 7 Apr 2021

Dear Prof. Craig O'Neill,

We sincerely thank you for your valuable input and constructive suggestions for this manuscript. We believe that most suggestions will help us improve our manuscript, as we are currently preparing it for re-submission. As a summary, we are currently incorporating your comments about the presentation of the results, the structure of the related text, sequence of the figures, background material that appears in the discussion section, and a more detailed discussion of some key model limitations. In particular, we

will better clarify the possible effects of our assumptions in terms of rheology and heat sources in the mantle, and we will enhance the discussion of our 2D approximation. Finally, we will move a subset of our results (i.e., model suite 1) to the supplementary material, also to make room for a better presentation of the more important model suite 2 in the main text.

Specific responses to each individual comment will be given when we re-submit our manuscript. As we are currently running a couple of additional models with much higher resolution (a thorough resolution test is requested by the other referee Prof. Shije Zhong), we expect our re-submission to be ready in a few weeks. We aim to address all comments raised by both reviewers, including the minor ones not mentioned above.

We appreciate your efforts and hope that, once our manuscript will be re-submitted, you will find that the revised manuscript will properly accommodate the points you raised.

Yours sincerely,

Anna Gülcher, Maxim Ballmer, and Paul Tackley.

---

## Author Comment (AC2) · 7 Apr 2021

Dear Prof. Shije Zhong,

We sincerely thank you for your valuable input and constructive suggestions for this manuscript. You raised some fair concerns regarding choices in our model set-up, and we are currently in the process of addressing these concerns to improve our manuscript for re-submission.

As a summary, we are improving our study and manuscript along the following lines: (i) we aim to address your main concern by including a substantial resolution test, and

discussing the effects of resolution on our estimates for heterogeneity preservation in the mantle; (ii) we will discuss in more detail the limitations of our models and choices made regarding the model set-up, and their possible influence upon our conclusions. In particular, these include the absence of internal heating (concern also raised by co-referee prof. Craig O'Neill), 2D vs. 3D geometry, and rheological parameters (both raised by co-referee); (iii) finally, we are currently re-structuring the presentation of our main results and supplementary material (raised by co-referee), to give priority to the models that are most relevant to Earth, and are primarily addressed in our Discussion (i.e., model suite 2).

As we are currently running high-resolution models as requested, it may take several weeks to finalize our re-submission. Specific responses to each individual comment from both reviewers will be given along with the resubmission of our manuscript. We aim to address all comments raised by both reviewers, including the minor ones not mentioned above.

We appreciate your efforts and hope that, once our manuscript is re-submitted, you will find that the revised manuscript will comply with the points you raised.

Yours sincerely,

Anna Gülcher, Maxim Ballmer, and Paul Tackley.

---

## Author Response (AR1)

**Reply to editor and reviewers**

Dear Julianne Dannberg; Craig O'Neill, and Shije Zhong,

Herewith we resubmit our revised manuscript entitled "Coupled dynamics and evolution of primordial and recycled heterogeneity in Earth's lower mantle" for *EGU: Solid Earth*. We thank both Craig O'Neill and Shije Zhong for their valuable input and constructive suggestions for this manuscript. They raised some fair concerns regarding the presentation of our results and clarifications of choices in our model set-up. We have addressed these concerns in this revised manuscript.

As a summary, we have improved our study and manuscript along the following lines: *(i)* we included a resolution test, and discuss the effects of resolution on our estimates for heterogeneity preservation in the mantle (major point by Shije Zhong); *(ii)* we discuss in more detail the limitations of our models and choices made regarding the model set-up, and their possible influence upon our conclusions. In particular, these include the absence of internal heating, 2D vs. 3D geometry, and rheological parameters (concerns raised by both reviewers); *(iii)* finally, we have re-structured the presentation of our main results and supplementary material (suggested by Craig O'Neill), to give priority to the models that are most relevant to Earth, and are primarily addressed in our Discussion (i.e., model suite 2).

Specific responses to each individual comment from both reviewers are given on the following pages. We also created a document in which all changed parts in the revised manuscript compared to the original one are highlighted (in blue). We appreciate your efforts and hope that you will find that the revised manuscript properly accommodates the  points raised, and is indeed suitable to feature in *EGU: Solid Earth*.

Yours sincerely,

Anna Gülcher, Maxim Ballmer, and Paul Tackley.

**Response to points raised by reviewer #1 – Craig O'Neill**

**Presentation of the results**

**Figure 2:** I feel like this figure should come after the figures showing the timeseries of behaviour, which probably feeds into the whole structure of the results. Show the models first, and describe the mixing patterns, and then present a regime diagram of mixing. Showing this first is topsy-turvy and throws the structure of this section off.

**Response:** In the Results section of our manuscript, we have re-ordered the presentation of the figures, and discussion thereof: first, the mixing patterns are presented and described (new manuscript Figures 2 and 3), and thereafter the regime diagram (new manuscript Figure 4) of mixing is discussed (see new subsections 3.1.1-3.1.5 and 3.2).

**Line 252-253:** If they are not directly related to the Earth, it does beg the question of whether you want to include them or not. When I read this part, it strikes me as student project (I know, I know, but best avoid that impression in a published paper), where you made them again and went back and had to do them again. I mean, that's how we work, but often a paper is improved by leaving out the bits that are not really relevant to the final message. Could these be an appendix?

**Response:** In response to, and agreement with, this concern, we have re-organised the presentation of our results in our manuscript. In particular, we have moved the first model suite (including the chemical layering regimes) to the appendix (D), as we mention in the manuscript that these experiments are not so relevant for Earth. By doing so, we created more space to focus on the experiments that are most relevant for the discussion and our final message (i.e., model suite with variable $D_{prim}$).

**Discussion of 2D vs 3D mixing**

**Line 50-54:** I note you mention 3D mixing in the discussion later, but I feel it is worthwhile bringing that content into the background here. In particular, Coltice and Schmalz showed the differences between 2 and 3D for high-Ra regimes were not that significant, so you could bring that discussion here. For more realistic convecting systems, you might considering looking at these 2:

O'Neill, C., Debaille, V. and Griffin, W., 2013. Deep earth recycling in the Hadean and constraints on surface tectonics. American Journal of Science, 313(9), pp.912-932.
https://www.ajsonline.org/content/313/9/912.short

O'Neill, C.J. and Zhang, S., 2018. Lateral mixing processes in the Hadean. Journal of Geophysical Research: Solid Earth, 123(8), pp.7074-7089.
https://agupubs.onlinelibrary.wiley.com/doi/full/10.1029/2018JB015698

Which I somewhat bashfully suggest as I was involved in them (thus - take it or leave it, but relevant if you want to justify using 2D c/w 3D).

**L483-487:** Again I think this whole discussion of 2D vs 3D is best nipped in the bud at the start - see previous comment.

**Response:** We have included a more elaborated discussion on 2D versus 3D mixing at the beginning of our discussion (lines 336-346), instead of at the very end of the paper. As 2D vs 3D mixing does not affect our problem statement or motivation, we did not include 2D vs 3D mixing already in the introduction (lines 50-54). In short, we expect that there may be two competing effects of 3D geometry on heterogeneity preservation: on one hand, coherent, viscous "blobs" may be difficult to preserve in the presence of toroidal and poloidal flow components; on the other hand, mantle flow may be even

more efficiently guided around viscous blobs in 3D than in 2D geometry. Indeed, it has been shown that sheet-like plumes in 2-D drive more efficient mixing than localized upwelling in 3-D convection models. We note that we are also currently running 3D models for future investigations.

**Discussion of thermal evolutionary aspects of models**

**L490:** Without radioactivity or core evolution, these aren't strictly evolutionary models, they are static steady-state models, started at adiabatic conditions, that are run for long periods of time. Apart from the destabilisation of heterogeneities, there is not much 'evolutionary' in them, and this point needs to be made clearer, and earlier.

**L508-510:** Again, this statement needs to be couched. The timescales for systems with significant radioactive decay are likely very different from the steady-state systems here. A CMB temperature of 4000K is probably ok for some of the early Earth
- it's a little higher than estimated today - but probably underestimates early evolution significantly. I think needs to be tackled more explicitly in the discussion, along the lines: "What can we tell about mixing timescales on the Earth from steady state models? Well, we select TCMB and adiabat values around ballpark (average for this period
- acknowledging these values evolved rapidly during this time). Based on this we can say..." I don't think this is too much overhead, but at least shuts down the critique of the evolutionary aspect missing here.

**Response:** We have included a more elaborate discussion on the choice of thermal boundary conditions and internal heating. First, we highlight these choices more specifically in the methods section (lines 84-86). Here, we note that although we do not incorporate internal heating or core cooling, our modelled internal mantle temperature does start off hot (potential temperature of 1900 K) and cools down to values considered appropriate for present-day Earth (potential temperature of ~1600 K).
Further, we have included a more detailed (and self-critical) discussion on the possible effects our choices, particularly in terms of early model evolution (lines 455-474).

**Minor points**

**Line 66-67:** "robustly predicted in many experiments" - better add a cite here.

**Response:** This sentence relates to our own models in the study presented in this paper.

**Line 69:** "a hybrid state" - do you think it is terribly different from marble cake with scale variance?

**Response:** If so, the scale variance would be multiple orders of magnitude, which we do not think agrees with "marble cake mantle" if there are viscous regions/blobs of several 100s of km large versus thin streaks on the km-scale.

**Line 76:** "512.96" looks like a decimal, expand to "512 x 96 cells"

**Response:** Typo corrected.

**Table 1.** The activation volume and energies are the same for the upper mantle (mostly olivine) and bridgmanite (PV). This is a bit of a stretch, and it was worth noting this in the review (although I'm sure nothing new to the senior authors). The energies themselves are very low compared to lab

measurements (c/w Hirth and Kohlstedt, or Karato and Wu for olivine, E_375e3), and the activation volume is appropriate for bridgmanite, but not for olivine. This certainly makes the models easier to run as things converge, but is it realistic? There is some token justification (lower mantle/grain size/stress) but this is not very well developed. I would like to see a little more text on justifying using these parameters properly, and what the potential implications are of this choice (eg. for the upper mantle, it results in very different behaviour near the lithosphere).

**Response:** We have included a bit more text on the justification of our chosen parameters in lines 113-120

**L147-148:** "density (FeO enrichment)": can you expand on the exact density difference between FeO and bridgmanite in the lower mantle, as this is quite relevant here. It sounds like you are saying FeO is denser - I encourage to revisit this statement carefully (and calculate the density profiles of bridgmanite vs FeO in burnman to see what I mean).

**Response:** We mean FeO enrichment of the bridgmanite (Mg# in $(Mg,Fe)SiO_3$), which would make the bridgmanitic material denser. We do not mean FeO versus bridgmanite. To clarify this, we have rewritten this to "intrinsic density (FeO-enrichment in $(Mg,Fe)SiO_3$)

**L264:** initial primordial layer thickness: can you justify this better? why 1844km thick?

**Response:** The range of layer thicknesses was chosen based on a test model suite. See section 2.4 and Appendix D.

**L285-287.** Could you expand on this more? I'm familiar with the work, so I get what you're saying, but not everyone will. Why exactly does a thinner bridgmanite layer lead to lower melting? Is it a fertility issue? Or dynamic? And maybe expand on why hotter mantle leads to more melting and thus a cooler mantle - those thinking in terms of plate cooling only will be thrown by this point.

**Response:** Rewritten the sentences (revised manuscript lines 275-281) to better clarify.

**L351-356:** Disproportionation of FeO during core formation is the most obvious (and increasingly commonly cited) cause of lower mantle heterogeneity. It's not explicitly mentioned here - I recommend Frost and McCammon (2008) for a summary.

**Response:** Disproportionation of FeO is now also mentioned here (revised manuscript lines 353-356)

**L362-364:** This goes off on a tangent here - is it needed? I feel this opens a can of worms you don't have space to really cover.

**Response:** the original lines here are disregarded in the revised manuscript. In fact, many of the first several subsections of the Discussion are completely rewritten to have a better flow.

**L375:** I think Rhodri thought he could explain the anticorrelation with the thermal (PPV) mechanism.

**Response:** as we rewrote the discussion section, this comment is not relevant anymore and the anticorrelation is not mentioned in the new version of the manuscript.

**L382-391:** The discussion in generally is a bit more disjointed than a lot of the rest of the paper, and skips from point to point without much continuity. The section in particular is hard to follow, and just really needs a rewrite.

**Response:** as mentioned above, many subsections of the Discussion, and this the first two sections in particular, are completely rewritten in the new version of the manuscript. We hope the reviewer will find that the flow is improved

**L408-411**: Can you explain better what the iron spin transition has to do with bridgmanite structure size/scale? This is not clear.

**Response:** rewritten and clarified in lines 382-389

**L432-434:** MgO-rich komatiites are very much more dense (at the surface) than basalt – are you talking about after the eclogite transition? If so, this statement needs to be. Much clearer, as written it is quite wrong. See van Thienen et al (2004) for effect of MgO content on mafic rocks.

**Response:** Due to the uncertainties of the exact composition of ancient basalt, and their densities at high pressures, we rewrote this sentence (new manuscript lines 412-414). Indeed, the iron contents of MORB and Archean basalts overlap with each other.

**L456-457:** I think you need to cite the old original Stein and Hoffman (1994) mantle overturn model here: Stein, M. and Hofmann, A.W., 1994. Mantle plumes and episodic crustal growth. Nature, 372(6501), pp.63-68.

**Response:** paper cited in line 438

**L477-479:** A disjunct with seismic models seems problematic. It's one thing to point it out, but as it stands you seem to be letting it undermine the whole work. You should really advance a reason for the discrepancy (are the differences – quantitatively – so large in terms of T? Is your core evolving? (No? Then it may overestimate heat flow. Etc etc).

**Response:** We do not agree that a possible "invisibility" of bridgmanite-enhanced regions in the mid-mantle in seismic tomography images is problematic at all. The prediction that bridgmanitic blobs in the lower mantle may not be visible in the lower mantle due to the competition of thermal and chemical effects on seismic velocity does NOT present a discrepancy with observations. It only highlights the necessity to think more carefully about how to test our models, perhaps in many different ways than just with seismic tomography. We hope this is now better clarified in the Discussion in lines 481-491. We note that we are currently working on quantitatively translating our models to seismic velocity models; so a quantitative study is on its way.

**Response to points raised by reviewer #2 – Shije Zhong**

**Main concern: resolution**
**3)** My main concern is about whether the 2-D models have sufficient numerical resolution to resolve the entrainment process that is so key to some of the model results reported here, e.g., the survival of the primordial materials and formation of the piles above the CMB. The 2-D models with fairly high Rayleigh number (_10ˆ7) have _100 grid points in radial direction covering _3000 km thick mantle. As a comparison, Leng and Zhong (2011) showed that using adaptive mesh refinement method, mesh resolution corresponding to 256 elements in vertical direction is needed to reproduce the entrainment rate by Davaille's analytical prediction for Ra=10ˆ5. We tried unsuccessfully for computing entrainment rate in models with variable viscosity. Zhong and Hager (2003) also used the marker chain method to determine the entrainment rate for a plume with fixed buoyancy on the top of a dense layer, demonstrating the requirement of numerical resolution. Van Keken et al., (1997) in their benchmark paper also showed the difficulty in computing entrainment rate. In Li et al. (2014), their 2-D models only cover the bottom of the mantle with super-high resolution (_3 km?) to study the recycled crustal material interacting with the basal layer. Based on these studies, I have a good reason to believe that the models here (high Ra and variable viscosity) may have over-predicted the entrainment, but I do not know how much it would affect the overall conclusions. It would be very helpful for the authors to pick a case for a resolution test. To this end, I can see the need for at least two calculations using 200 and 400 grid points in radial direction, given the high Ra and variable viscosity in their models (both making entrainment more difficult to compute).

**Response**: We have included a resolution test for three selected models as an appendix (appendix B, lines 545-567, Figure B1) to investigate the effect of resolution on the numerical results, in particular in terms of heterogeneity preservation predictions. We observed that the heterogeneity estimates ($Xsi_{prim}$ and $Xsi_{basalt}$) are indeed slightly underestimated in our main models with a resolution of 512x64 (i.e., compared to higher-resolution models). That said, we did *not* observe any significant changes in mantle dynamics or heterogeneity style prediction for all models explored in this resolution test (see details in manuscript).
Overall, we are convinced our main conclusions are robust, and model predictions are conservative in terms of the preservation of primordial material. We also note that we have run models with a 2048 times 384 grid resolution, but computational limitations only allowed these models to run until ~1.5 Gyr, and therefore we did not include these models in the supplement. For the first 1.5 Gyr model time there have been no significant differences between cases with 2084x384 and 1024x192 resolution.

**Radioactive heating:**
**2)** The manuscript acknowledged in the final sub-section the drawback of ignoring the radioactive heating. For a study like the current one aiming at the long-term thermal and chemical evolution of the mantle, this seems to be a significant deficiency. I think that the authors should acknowledge it in the abstract and conclusions sections.

**Response:** Reviewer #1 expressed similar concerns (see comment and reply above), and we have implemented better clarifications and discussions on our choices of thermal parameters in several parts of the revised manuscript: lines 84-86 (methods) and 456-475 (discussion). We respectfully disagree that the draw-back of ignoring radioactive heating should be mentioned in the abstract already, as usually limitations are not already mentioned here.

**Clarification on viscosity:**
**3)** It would be helpful to clarify how the viscosity is computed. Equation (1) gives the viscosity for each composition through the pre-factor, lambda. I suppose that in solving the Stokes' equation, the viscosity

is assigned to grid points. Then how are viscosities of different composition in a vicinity of a grid point (or within in a grid) averaged and assigned to the grid?

**Response:** Indeed, composition is stored on tracers, yet viscosity is calculated on the grid. Composition (in either harzburgite-basalt or primordial space) is averaged to the grid in a standard way: averaging nearby tracers weighted by their mass and distance ("shape function averaging": Tackley and King, 2003, DOI: 10.1029/2001GC000214). In case the cell contains a mixture of compositions, the overall cell viscosity of calculated as a mass-weighted geometrical average of the viscosities of different compositions. We have improved the explanation of how tracer properties are interpolated to the grid by amending the text below equation (1) to clarify this (lines 110-112).

**4)** To follow point 3 above on viscosity, it seems that the manuscript focuses on the effect of lambda_prim, while not mentioning much on lambda_LM and lambda_ppv. Also, it appears that lambda_prim controls whether a convection is in a stagnant lid (for a small lambda_prim) regime or mobile-lid regime (for a large lambda_prim). Can the authors provide more insight or explanation as to why lambda_prim could have such an effect? Most previous studies show that for this type of models, large activation energy would lead to stagnant-lid convection, but the current study seems to suggest otherwise. It is unclear to me why lambda_prim has such a power. Along this line, does a large lambda_prim mean a more viscous blob (i.e., high viscosity blob)? Here a high radioactive heating for the primordial material may potentially make a big difference, as it would heat up the blob over time.

**Response**: lambda_prim and lambda_ppv are in fact types of the compositional pre-factor in the viscosity equation (1). We have included this clarification in the methods section lines 160-162, to make sure the readers understand this. As lambda_LM and lambda_ppv are kept constant in all models, they are not the target discussion points of our paper. Lambda_prim is indeed one of the key parameters in our study, and we also systematically change this parameter between models (see Section 2.4). It stands for the intrinsic viscosity contrast between the primordial material and all other materials, i.e., the intrinsic strength of the primordial material (motivated by the intrinsic strength of bridgmanite). So yes, a large lambda_prim means a more viscous blob, as is stated in section 3.2. As Lambda_prim directly controls heterogeneity preservation (Figures 2-4), mantle layering, and mantle viscosity profile, it also has a secondary effect on tectonic style. We now clarify this in more detail in lines 110-112, . As for the possible effect of internal heating, see line456-475. We note that it is likely that bridgmanitic domains are depleted in internal heat sources 471-475).

**Other comments:**

**L168** Regime I reported in Gulcher et al., (2020b) does not exist anymore in the current study, as stated in line 168, and can the authors explain why Regime I does not exist in the current models?

**Response:** this is now clarified at the beginning of the Discussion in lines 319-324

**Fig. 7**, what prevents the primordial materials from entering the upper mantle? The phase change or compositional buoyancy or both? Some explanation would be helpful.

**Response:** Primordial material that is entrained/eroded as small blobs from the larger bridgmaitic domains enters the upper mantle without any specific restrictions. However, this process happens very slowly because erosion rates are low. Bridgmanitic materials are not significantly restricted [i.e., likely less restricted than pyrolitic plume-like upwellings] by the 660 phase change because they are not very hot. Big primordial blobs tend to be preserved in the cores of convection cells. Convection patterns are further stabilized by this configuration. Along these lines, significant amounts of primordial material

can survive in the mid mantle. We have further clarified this issue at the description of the relevant rgeime (see revised manuscript section 3.1.2)

**L441-443,** " : : : ongoing debate about whether thermochemical piles are intrinsically stable : : :". More appropriate references may be McNamara and Zhong (2005, Nature) and Zhang et al., (2010, JGR).

**Response:** papers included in the references here.

**Technical Comments:**
**Line 35**, delete "to".

**Response:** Done.

**Line 178**, Fig. 4a should 3a.

**Response:** We have corrected this.

**Fig. 3**'s caption, "regime I" or regime II? Fig. 4's caption, "regime II" or regime III? In general, the regime names are somewhat confusing. Another example is in lines 200 (the title of subsection 3.1.2) and 201: the sub-title in line 200 says "regime III", but line 201 says "2nd regime".

**Response:** These were indeed inconsistencies (typos). We thank the reviewer for pointing this out. We have fixed all occurrences in the text.

**Fig. 4a**, B=64 or 0.64?

**Response:** This was indeed a mistake, which is now corrected. We thank the reviewer for pointing this out.

**Fig. 10**, BEAMS? What does it stand for?

**Response:** We have now included what BEAMS stands for in the caption of the figure.

---

## Author Response (AR2)

**Reply to editor and reviewers**

Dear Juliane Dannberg and Shije Zhong,

Herewith we resubmit our revised manuscript entitled "Coupled dynamics and evolution of primordial and recycled heterogeneity in Earth's lower mantle" for *EGU: Solid Earth*. We thank Shije Zhong for his final comments regarding our resolution tests. We have addressed the two points raised by the reviewer in the final version of our manuscript and in the detailed responses below. Moreover, in this final manuscript, we have decided to move our extensive Appendix to a separate file (Supplement), to be downloaded separately, rather than being attached to the main paper as a very long Appendix.

We appreciate your efforts and hope that you will find that the revised manuscript properly accommodates the two points raised, and is suitable to feature in *EGU: Solid Earth*.

Yours sincerely,

Anna Gülcher, Maxim Ballmer, and Paul Tackley.

**Response to points raised by reviewer #2 – Shije Zhong**

The authors made significant effort in revising the manuscript and I am satisfied with all the responses except that I think that the response to resolution issue needs some more clarification. The newly added appendix B on resolution tests is helpful. The authors used 3 diagnoses for resolution tests: relative volumes of primitive material and ROC in the lower mantle and RMS velocity, v_RMS (Fig. B1). The authors concluded that their results are qualitatively unchanged for different resolutions, which I agree and was what I suspected in my original review.
**However, two issues need to be clarified and acknowledged:**

**1)** v_RMS (Fig. B1c) shows some modest resolution dependence even at the highest resolution (~15% change when the resolution is doubled, which is not insignificant, in my view).

**Response**: thank you for pointing this out. Indeed, the largest v_RMS change between 512x96 and 1024x192 resolution models is approximately -11% for the models MdD30 and MdD300. Note that for our reference model MdD100 (black icons), this is not the case. We have added a few sentences highlighting the v_RMS trends in lines 46-48 in the Supplement, to avoid any misunderstanding that v_RMS is completely insensitive to resolution.

**2)** The relative volumes in the lower mantle are different from entrainment which often measures the change of mass or volume for a chemical reservoir or domain with time in previous studies (e.g.,

Zhong and Hager, 2003) and is probably more sensitive to resolution. Perhaps, in using the relative volumes in the lower mantle, the small drips of entrained materials are also included, thus making relative volumes in the lower mantle less sensitive to resolution. An extreme case is to compute the relative volume for the whole mantle, in which case the relative volume is a measure of the total conservation of composition. I am sorry if I sound somewhat insistent on this issue, as I have seen over the years that the calculations of entrainment or preservation of chemical reservoirs (e.g., LLSVP) in numerical models are over-simplified. I think that it would be helpful for the authors to acknowledge these two points in the paper to avoid misunderstanding.

**Response**: We respectfully insist that the quantities currently shown (vol% of chemical reservoirs Xsi_LM_prim and Xsi_LM_bs in the lower mantle are key quantities to show for several reasons. First, these parameters are extensively used in the main text and Figures and they could, for example, be used for comparisons. We also link these quantities with the different styles of heterogeneity preservations throughout the manuscript. Moreover, in our numerical models, it is difficult to quantify entrainment rates in the mantle in the same way as is done in e.g. Zhong and Hager, 2003, or really in any way (see below). In our models, the bulk composition of the convecting mantle changes as a function of time due to upper-mantle processing (i.e., melting) of primordial material, technically reflected as a conversion of a tracer from primordial to harzburgite-basalt space (see Methods section). Therefore, there is no true "conservation of total composition" and entrainment rate as calculated in other papers is not straightforward.

Primordial material is entrained by the convecting mantle at the margins of blobs with high primordial-material content, but then remains floating in the mantle for several cycles before being processed. After having made several tests (inspired by the reviewer's comment), we conclude that the Xsi_LM_prim is an appropriate quantity that reflects the integrated entrainment of primordial material over time.

Similarly, basaltic materials are entrained from thermochemical piles in the lower mantle, but they also keep on being added to the piles by segregation of basalt from harzburgite. Again, it is difficult to isolate the effects of entrainment of basalt in our models without detailed analysis that goes far beyond the scope of the resolution test. Nevertheless, Xsi_LM_bs is a relevant quantity that reflects both entrainment and segregation of basalt over time. Both are expected to be resolution dependent, but we demonstrate that the net effect does not change our conclusions.

We now better clarify this in the Supplement, and have added additional lines #55-58 and #60-62 of the Supplement. We hope you agree with our statements.

---

## Author Response (AR3)

**Reply to executive editor**

Dear Susanne Buiter,

Herewith we resubmit our revised manuscript entitled "Coupled dynamics and evolution of primordial and recycled heterogeneity in Earth's lower mantle" for *EGU: Solid Earth*. We thank your for mentioning the text overlap problem with our 2020 EPSL paper. It seems like we missed this comment made during the first review round, and we sincerely apologize for that. In the current manuscript version, we have significantly reworded (and restructured) the Methods section, in particular the paragraphs we identified to be very similar to our 2020 EPSL paper. We have attached two PDFs (one for the main paper and one for the supplement) in which all the changed text, relative to the previous manuscript version, is highlighted in blue.

We appreciate your efforts and hope that you will find that the revised manuscript properly accommodates the  problem raised, and is suitable to feature in *EGU: Solid Earth*.

Yours sincerely,

Anna Gülcher, Maxim Ballmer, and Paul Tackley.